# Search and Refine During Think: Facilitating Knowledge Refinement for Improved Retrieval-Augmented Reasoning

**Yaorui Shi[1*], Sihang Li[1*], Chang Wu[1], Zhiyuan Liu[2], Junfeng Fang[2],**
**Hengxing Cai[3†], An Zhang[1], Xiang Wang[1†],**

[1] University of Science and Technology of China
[2] National University of Singapore
[3] DP Technology
{yaoruishi, sihang0520, xiangwang1223}@gmail.com, caihengxing@dp.tech
[*] Equal contribution. [†] Corresponding author.

## Abstract

Large language models have demonstrated impressive reasoning capabilities but are inherently limited by their knowledge reservoir. Retrieval-augmented reasoning mitigates this limitation by allowing LLMs to query external resources, but existing methods often retrieve irrelevant or noisy information, hindering accurate reasoning. In this paper, we propose AutoRefine, a reinforcement learning post-training framework that adopts a new "search-and-refine-during-think" paradigm. AutoRefine introduces explicit knowledge refinement steps between successive search calls, enabling the model to iteratively filter, distill, and organize evidence before generating an answer. Furthermore, we incorporate tailored retrieval-specific rewards alongside answer correctness rewards using group relative policy optimization. Experiments on single-hop and multi-hop QA benchmarks demonstrate that AutoRefine significantly outperforms existing approaches, particularly in complex, multi-hop reasoning scenarios. Detailed analysis shows that AutoRefine issues frequent, higher-quality searches and synthesizes evidence effectively. Code is available at https://github.com/syr-cn/AutoRefine.

## 1 Introduction

Large language models (LLMs) have shown impressive abilities in language understanding, planning, and problem solving [1–3]. Recent advances demonstrate that reinforcement learning (RL) [4] further enhances LLMs' reasoning capabilities [5, 6], especially in complex tasks such as mathematics and coding [7, 8]. However, the knowledge encoded in LLMs is inherently constrained by their training corpora, limiting their reasoning performance on tasks requiring up-to-date information [9, 10].

A common strategy to address this limitation is retrieval-augmented generation (RAG), which equips LLMs with retrieval tools to access external knowledge bases during question answering [11–13]. Widely-adopted RAG pipelines typically rely on supervised fine-tuning (SFT) to train LLMs to issue search queries and generate responses based on retrieved documents [14–16]. While SFT can be effective for training large models for search, it sometimes necessitates the construction of high-quality search paths, which incurs additional effort and resource overheads [17]. To address this, recent studies draw inspiration from RL-based post-training [5] and explore RL for retrieval-augmented reasoning, achieving excellent results by only evaluating final answer correctness without the need for

39th Conference on Neural Information Processing Systems (NeurIPS 2025).

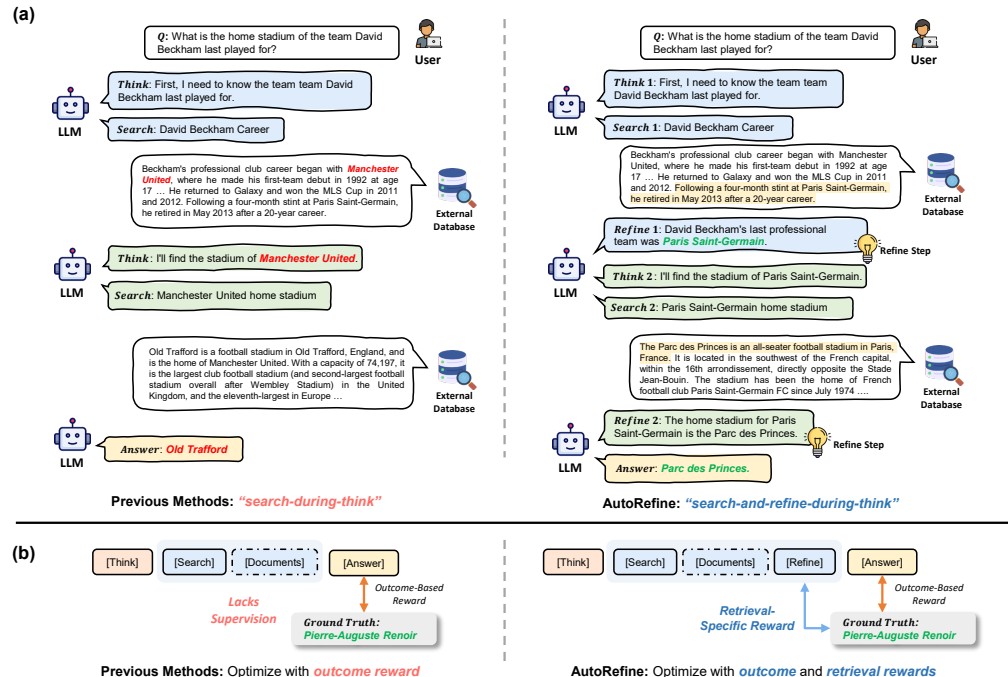

Figure 1: Comparison between previous retrieval-augmented reasoning methods and AutoRefine. (a) Previous "search-during-think" models can get distracted by irrelevant details between retrieval steps, leading to an incorrect answer. AutoRefine introduces a <Refine> step where the model explicitly refines crucial evidence, enabling the model to link information across multiple hops, plan its next query, and arrive at the correct final answer. (b) While previous methods rely only on outcome-based rewards, AutoRefine incorporates a retrieval-specific reward to directly supervise the <Refine> step.

pre-collected reasoning paths [18–23]. Scrutinizing existing studies on retrieval-augmented reasoning, we summarize a common "search-during-think" paradigm: given prompts with special search tokens (*e.g.,* '<search> ... </search>'), the LLM is trained via RL to autonomously invoke retrieval tools, retrieve some documents from external knowledge bases, and generate answers within '<answer> ... </answer>' using the retrieved information.

Despite their promising results, we identify two core limitations inherent in the current retrieval-augmented reasoning paradigm:

- **Lack refinement of retrieved documents.** When facing out-of-scope questions, LLMs often require pieces of precise factual information (*e.g.,* names of historical figures, dates of events). However, the current "search-during-think" paradigm typically uses retrieval tools to return full documents based on input queries, many of which are noisy or only weakly relevant. As illustrated in Figure 1(a), previous methods reason directly over raw retrieved content, making it susceptible to getting distracted by irrelevant details. This is particularly problematic in multi-hop scenarios, where a distraction in an early step can derail the entire reasoning chain.

- **Underexplored retrieval-specific rewards.** While prior work on RL post-training highlights the importance of reward design [24–26], most retrieval-augmented reasoning methods rely solely on an outcome-based reward — typically assessing the correctness of the final answer. As shown in Figure 1(b), this coarse supervision underexplores retrieval-specific rewards and offers little direct guidance for improving the retrieval process itself. As a result, it could be difficult for the LLM to learn how to retrieve more relevant or informative documents.

To address these limitations, we propose **AutoRefine**, a simple yet effective RL post-training framework that enhances the LLM's autonomous retrieval-augmented reasoning capability. At its core, AutoRefine adopts a "search-and-refine-during-think" paradigm, guided by a combination of answer and retrieval rewards. First, unlike prior "search-during-think" approaches that overlook refinement, we introduce an explicit knowledge refinement step into the reasoning loop using a '<search> ... </search>[documents]<refine> ... </refine>' template (§2.1). This template encourages the model

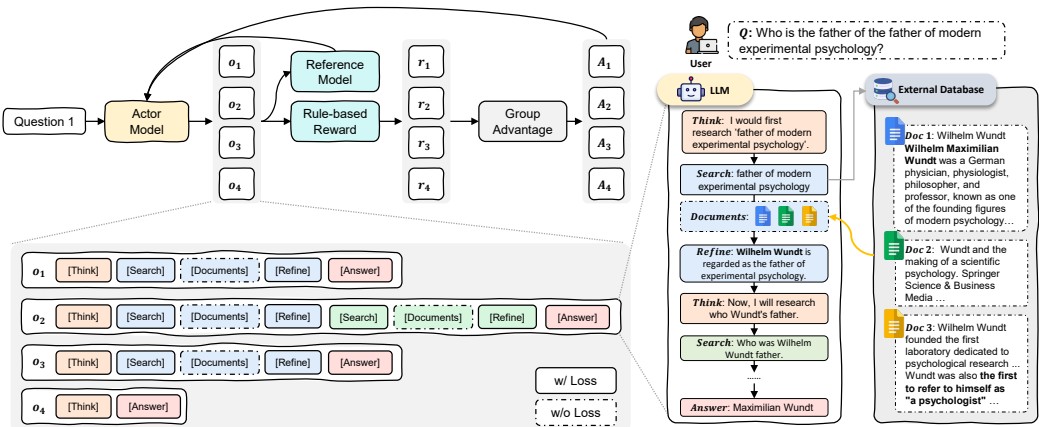

Figure 2: The training scheme of AutoRefine. (right) An actor model generates diverse reasoning trajectories for a given question, including think, search, refine, and answer. (left) These trajectories are optimized using the GRPO [7] algorithm described in Equation (4), where the loss on retrieved documents is masked out. We take $G = 4$ in this example.

to explicitly distill crucial evidence from retrieved documents. By isolating key facts, the model can better link information across multiple retrieval hops and accurately plan subsequent queries before generating a final answer. Second, under this template, we apply Group Relative Policy Optimization (GRPO) [5] to train the model with both outcome-based and retrieval-specific rewards, rather than outcome-based rewards alone (§2.2). Specifically, during training, we first sample several trajectories from the model, each consisting of a sequence of <think>, <search>, <refine>, and <answer> steps. While the answer reward evaluates the final output, the retrieval reward is computed based on the quality of the content within the <refine> blocks, providing direct supervision for the refinement step. This joint reward design explicitly guides the model to extract, organize, and utilize fine-grained knowledge throughout reasoning.

To empirically assess AutoRefine, we conduct experiments on both single-hop [27–29] and multi-hop [30–33] question answering (QA) benchmarks. AutoRefine surpasses leading methods [5, 18, 21, 19] by $6.9\%$ higher average accuracy, and shows especially high performances in multi-hop scenarios (*cf.* Table 1). It demonstrates a strong ability in identifying and addressing knowledge gaps via multi-turn, high-quality search queries. The knowledge refinement steps also effectively extract crucial information from noisy retrieved documents, directly contributing to improved answer quality. Additional experiments confirm the contribution of both the retrieval-specific reward and the refinement module, and AutoRefine holds robust performance across different retrieval depths.

## 2 Method

In this section, we introduce AutoRefine, a simple yet effective RL framework that enhances the LLM's autonomous retrieval-augmented reasoning capability. We first outline the overall task formulation and trajectory generation steps, highlighting the novel "search-and-refine-during-think" paradigm (§2.1). Next, we detail our reward modeling with both answer-based and retrieval-specific rewards to encourage fine-grained knowledge refinement (§2.2). Due to limited space, we refer to Appendix A for related work about reasoning in LLMs and retrieval augmented generation.

### 2.1 Trajectory Generation with Searching and Refinement

**Task Formulation.** Given a dataset $\mathcal{D} = \{(q, a)\}$ containing question–answer pairs and an external search engine $\mathcal{E}$, the task of retrieval-augmented reasoning requires the LLM to generate reasoning trajectories $o$ by iteratively interacting with the knowledge source $\mathcal{E}$. Formally, for each question $q$, we generate a reasoning trajectory: $o = (\tau_1, \tau_2, \ldots, \tau_T)$, where the $t$-th intermediate reasoning step $\tau_t = (s_t, c_t)$ consists of an action $s_t \in \{\texttt{<think>}, \texttt{<search>}, \texttt{<documents>}, \texttt{<refine>}, \texttt{<answer>}\}$ and its associated content $c_t$. The model is expected to repeatedly retrieve and refine knowledge from $\mathcal{E}$ until reaching a final answer $o_{\text{ans}}$ that correctly addresses the question $q$.

> You are a helpful assistant who is good at answering questions with multi-turn search engine calling. To answer questions, you must first reason through the available information using <think> and </think>. If you identify missing knowledge, you may issue a search request using <search> query </search> at any time. The retrieval system will provide you with the three most relevant documents enclosed in <documents> and </documents>. **After each search, you need to summarize and refine the existing documents in `<refine>` and `</refine>`.** You may send multiple search requests if needed. Once you have sufficient information, provide a concise final answer using <answer> and </answer>.
> <user> Question: {**QUESTION**} </user>

Figure 3: Prompt template for rollout generation.

**Rollout Generation.** The actor LLM $\pi_\theta$ generates trajectories by performing multiple rounds of interactions with the search engine $\mathcal{E}$. The trajectories contain multiple internal reasoning cycles, as illustrated in Figure 2. Each cycle consists of a sequence of structured operations: "<think>...</think>" for overall planning of consequent search actions, "<search>...</search>" for querying the external search engine, "<document>...</document>" for incorporating the retrieved documents, and "<refine>...</refine>" for distilling relevant information from the retrieved content. Following the reasoning phase, the model generates the final response within the "<answer>...</answer>" block based on the refined knowledge. Notably, the number of internal cycles is not manually pre-defined but autonomously determined by the actor LLM, adapting dynamically to the difficulty of the question. These tokens are defined and explained to the model via system instructions (*cf.* Figure 3).

**Stopping Criteria.** The generation terminates when an answer action is produced, *i.e.,* $s_T = $ <answer>. The content $c_T$ of the terminal state $\tau_T$ is extracted as the trajectory's final answer $o_{\textbf{ans}}$.

## 2.2 Reward Modeling with Retrieval-Aware Signals

We use simple rule-based rewards to encourage free exploration during the RL process. The reward in AutoRefine consists of two complementary components: (1) the *Outcome-Based Reward*, which directly assesses the correctness of the answer generated by the model, and (2) the *Retrieval-Specific Reward*, which encourages the model to accurately identify and extract answer-relevant information from the retrieved documents.

**Outcome-Based Reward.** The outcome-based reward (*aka.* the answer reward) $\mathcal{R}_{\text{Ans}} \in [0, 1]$ compares the model's final answer within the <answer></answer> block to the ground-truth answer, measuring its correctness. Formally, we treat the predicted and ground-truth answers as sets of words, and use F1-score between these two sets as the reward:

$$\mathcal{R}_{\text{Ans}} = \text{F1}(o_{\text{ans}}, a) = \frac{2|o_{\text{ans}} \cap a|}{|o_{\text{ans}}| + |a|}, \tag{1}$$

where $o_{\text{ans}}$ is the predicted answer (*e.g.,* "Pierre-Auguste Renoir" in yellow box of Figure 1(a), and $a$ is the ground truth answer from the $(q, a)$ pair (*e.g.,* the gray box in Figure 1(b)).

**Retrieval-Specific Reward.** We further introduce an additional reward, the retrieval reward $\mathcal{R}_{\text{Ret}} \in \{0, 1\}$, to explicitly encourage the extraction and utilization of relevant information from noisy retrieved documents. The retrieval reward is measured based on the quality of refined documents within the <refine></refine> blocks. Specifically, we collect all knowledge refinement steps (*i.e.,* content within the <refine>...</refine> blocks) across the trajectory and concatenate them into a single text sequence:

$$\mathcal{R}_{\text{Ret}} = \mathbb{I}(a \cap o_{\text{refine}} = a), \tag{2}$$

where $\mathbb{I}(\cdot)$ is the indicator function, $o_{\text{refine}} = \bigcup \{ c_t \mid (s_t, c_t) \in o \land s_t = \texttt{<refine>} \}$ is the concatenation of all the knowledge refinement steps (*e.g.,* "The documents concludes...'The Umbrellas'." in blue box of Figure 1). This reward is activated when all components of the ground-truth answer are present in the refined knowledge, reinforcing faithful and targeted information extraction.

**Integrating Outcome and Retrieval Rewards.** The overall reward function in AutoRefine is designed to encourage both accurate final answers and meaningful intermediate knowledge extraction. Specifically, the model receives a full reward of $1$ if it generates the correct answer. If the final answer

is incorrect but some relevant information has been extracted during the refinement step, a partial reward of 0.1 is assigned. No reward is granted if neither correct answers nor relevant information are produced. The overall reward $\mathcal{R}_{\text{Overall}}$ can be formally written as:

$$\mathcal{R}_{\text{Overall}} = \begin{cases} \mathcal{R}_{\text{Ans}}, & \text{if } \mathcal{R}_{\text{Ans}} > 0 \\ 0.1, & \text{if } \mathcal{R}_{\text{Ans}} = 0 \text{ and } \mathcal{R}_{\text{Ret}} > 0 \\ 0. & \text{if } \mathcal{R}_{\text{Ans}} = \mathcal{R}_{\text{Ret}} = 0 \end{cases} \tag{3}$$

**Training Objective.** We apply Group Relative Policy Optimization (GRPO) [7] as the policy optimization algorithm for RL. The overview of the GRPO training scheme is shown in the top-left corner of Figure 2. Formally, given an actor model $\pi_\theta$ and a reference model $\pi_{\text{ref}}$, a group of $G$ rollouts $\{o_i\}_{i=1}^{G}$ is sampled as described in §2.1. We optimize the actor model $\pi_\theta$ by maximizing:

$$\underset{\theta}{\arg\max} \; J_{\text{GRPO}}(\theta) = \mathbb{E}_{(q,a)\sim\mathcal{D},\{o_i\}_{i=1}^{G}\sim\pi_{\theta_{\text{old}}}(\cdot|q)} \left[ \frac{1}{G}\sum_{i=1}^{G}\frac{1}{|o_i|}\sum_{t=1}^{|o_i|} \min\left( \frac{\pi_\theta(o_{i,t}\mid q, o_{i,<t})}{\pi_{\theta_{\text{old}}}(o_{i,t}\mid q, o_{i,<t})}\hat{A}_{i,t}, \right. \right.$$
$$\left. \left. \text{clip}\left( \frac{\pi_\theta(o_{i,t}\mid q, o_{i,<t})}{\pi_{\theta_{\text{old}}}(o_{i,t}\mid q, o_{i,<t})}, 1-\epsilon, 1+\epsilon \right)\hat{A}_{i,t} \right) - \beta\mathbb{D}_{\text{KL}}\left[\pi_\theta \| \pi_{\text{ref}}\right] \right]$$
$$\tag{4}$$

where $\hat{A}_{i,t} = [r_{i,t} - \text{mean}(r_t)]/\text{std}(r_t)$ is the normalized token-level advantage for the $i$-th rollout in the group, $G$ is the group size, $\epsilon$ is the clipping ratio, and $\beta$ is the coefficient for the estimated KL divergence. As shown in the bottom-left part of Figure 2, we mask out the retrieved documents during the loss computation.

## 3 Experiments

In this section, we aim to answer the following Research Questions (RQs):

**RQ1**: How effectively does AutoRefine's "search-and-refine-during-think" paradigm enhance performance in retrieval-augmented question answering?

**RQ2**: Can AutoRefine effectively resolve information gaps through retrieval, especially when facing complex multi-hop problems?

**RQ3**: Can knowledge refinement steps distill critical information from retrieved documents?

**RQ4**: Can AutoRefine achieve robust performance improvements under different retriever settings?

### 3.1 Experiment Setup

**Datasets.** We evaluate performance using seven diverse QA benchmarks, including three single-hop QA datasets: Natural Questions (NQ) [27], TriviaQA [29], PopQA [28], and four datasets that require multi-hop searching: HotpotQA [30], 2WikiMultihopQA (2Wiki) [31], Musique [32], Bamboogle [33] for evaluation. Exact match accuracy serves as the evaluation metric for all downstream datasets. Following the setting of prior works [19], we train AutoRefine using a combined training set from NQ and HotpotQA.

**Baselines.** In our experiments, we compare AutoRefine against three kinds of methods: (1) generation without retrieval (w/o Retrieval), including direct generation with LLM, supervised fine-tuning (SFT), and R1-like training (R1) [5] without Retrieval; (2) methods with single-hop retrieval (w/ Single-Hop Retrieval), including direct retrieval with the input question (Naive RAG); (3) training with retrieval (w/ Multi-Hop Retrieval) including: agentic search method Search-o1 [18], IRCoT [34], retrieval-augmented reasoning model Search-R1 [19] and ReSearch [21].

**Implementation Details.** To simulate a real-world search scenario, we remove original context documents from the QA datasets [30–32] and instead use the December 2018 Wikipedia dump [35] as the external knowledge source, with E5-base-v2 [36] as the retrieval engine. By default, the search engine retrieves the top three most relevant documents on each query. For RL-based baselines, we run experiments using both Qwen2.5-3B-Base and -Instruct models. For SFT and direct generation

Table 1: (**RQ1**) Accuracy comparison of AutoRefine versus baseline methods with Qwen2.5-3B [3] across various QA benchmarks. **Bold** denotes best results, and underline denotes second best results.

| Methods | Single-Hop QA | | | Multi-Hop QA | | | | |
| --- | --- | --- | --- | --- | --- | --- | --- | --- |
| | NQ | TriviaQA | PopQA | HotpotQA | 2Wiki | Musique | Bamboogle | Avg. |
| w/o Retreival | | | | | | | | |
|    Direct Generation | 0.106 | 0.288 | 0.108 | 0.149 | 0.244 | 0.020 | 0.024 | 0.134 |
|    SFT | 0.249 | 0.292 | 0.104 | 0.186 | 0.248 | 0.044 | 0.112 | 0.176 |
|    R1-Instruct [5] | 0.210 | 0.449 | 0.171 | 0.208 | 0.275 | 0.060 | 0.192 | 0.224 |
|    R1-Base [5] | 0.226 | 0.455 | 0.173 | 0.201 | 0.268 | 0.055 | 0.224 | 0.229 |
| w/ Single-Hop Retrieval | | | | | | | | |
|    Naive RAG [37] | 0.348 | 0.544 | 0.387 | 0.255 | 0.226 | 0.047 | 0.080 | 0.270 |
| w/ Multi-Hop Retrieval | | | | | | | | |
|    Search-o1 [18] | 0.238 | 0.472 | 0.262 | 0.221 | 0.218 | 0.054 | 0.320 | 0.255 |
|    IRCoT [34] | 0.111 | 0.312 | 0.200 | 0.164 | 0.171 | 0.067 | 0.240 | 0.181 |
|    ReSearch-Instruct [21] | 0.365 | 0.571 | 0.395 | 0.351 | 0.272 | 0.095 | 0.266 | 0.331 |
|    ReSearch-Base [21] | 0.427 | 0.597 | 0.430 | 0.305 | 0.272 | 0.074 | 0.128 | 0.319 |
|    Search-R1-Instruct [19] | 0.397 | 0.565 | 0.391 | 0.331 | 0.310 | 0.124 | 0.232 | 0.336 |
|    Search-R1-Base [19] | 0.421 | 0.583 | 0.413 | 0.297 | 0.274 | 0.066 | 0.128 | 0.312 |
|    AutoRefine-Instruct | 0.436 | 0.597 | 0.447 | 0.404 | 0.380 | **0.169** | 0.336 | 0.396 |
|    AutoRefine-Base | **0.467** | **0.620** | **0.450** | **0.405** | **0.393** | 0.157 | **0.344** | **0.405** |

baselines, we use the instruct variant to better align with instruction-following tasks. Most baseline results are taken from Search-R1 [19], which has experimental settings consistent with ours. We reproduce ReSearch using the authors' publicly available code. Additional implementation details can be found in Appendix B, and further experimental results are provided in Appendix C.

## 3.2 Overall Performance (RQ1)

Table 1 presents the overall performance comparison between AutoRefine and the baseline methods. The Avg. column stands for the average accuracy. As shown in the results, AutoRefine significantly outperforms baseline models across the seven benchmarks. It achieves a $0.069$ accuracy gain on the base variant and a $0.060$ improvement on the instruct one compared to the strongest baseline.

Besides the overall performance, we observe that the performance gains achieved by AutoRefine are more obvious on the multi-hop QA benchmarks. For example, AutoRefine improves the performance on 2Wiki by $0.083$ and Musique by $0.045$, which implies $21\%$ and $26.7\%$ relative increase, respectively. According to further analysis in §3.3.1 and §3.3.2, we attribute AutoRefine's extraordinary performance on multi-hop benchmarks to its ability to perform high-quality searching and efficient utilization of retrieved documents.

> **Obs 1:** AutoRefine significantly improves QA accuracies, especially on multi-hop benchmarks.

## 3.3 Analytical Results

### 3.3.1 Search Behaviors (RQ2)

A crucial capability of retrieval-augmented reasoning models is identifying and addressing knowledge gaps via retrieval. To evaluate this, we analyze the **search frequency** and **search quality** of AutoRefine in four scenarios: training samples, all seven downstream benchmarks, single-hop QA benchmarks, and multi-hop QA benchmarks. These analyses are depicted in Figure 4.

**Search Frequency** reflects the model's capability to recognize knowledge gaps and perform searches accordingly. This search behavior is measured by the average number of search calls a model makes per rollout. We analyze the search frequency of both -Base and -Instruct variants of AutoRefine in Figure 4(a). As the figure shows, both variants have evolved multi-turn searching abilities after enough training steps. The average number of search calls converges to around $1.5$ for AutoRefine-Instruct, and higher than $2$ for AutoRefine-Base.

Despite the high overall search frequency on seven benchmarks, AutoRefine demonstrates distinct search behaviors for single-hop and multi-hop questions. On the three single-hop benchmarks, both

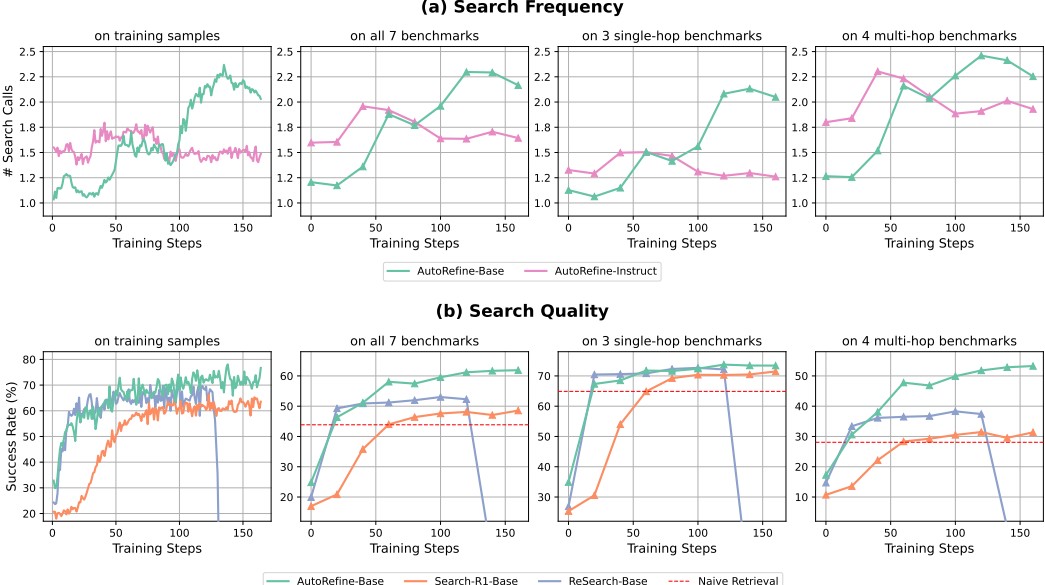

Figure 4: **(RQ2)** Visualization of the search behaviors. (a) AutoRefine's average number of search calls per rollout. For both variants, AutoRefine learns to adaptively issue more search queries for multi-hop questions and fewer for single-hop ones. (b) Comparison of search success rates between retrieval-augmented reasoning methods. While all methods draft more efficient search queries than naive retrieval, AutoRefine achieves more significant performance gains.

variants begin with fewer than $1.3$ searches per rollout, and gradually adjust to $2.0$ and $1.2$. In contrast, the models begin with much higher search frequencies when facing multi-hop questions, which rapidly go up to $2.0 \sim 2.5$. This phenomenon exhibits AutoRefine's ability to dynamically adjust the number of search calls according to the complexity of downstream tasks, with more frequent searching on multi-hop questions and less on single-hop ones.

> **Obs 2.1:** AutoRefine learns to perform multi-turn searching and can adaptively issue search queries depending on task complexity.

**Search Quality** evaluates whether the model generates effective search queries that can return informative documents. Knowledge-intensive questions often demand precise factual information to ask, *e.g.,* names of historical figures or dates of events. In such cases, search calls can only be considered successful if the retrieved documents directly contain the answer. Hence, we estimate the search quality by counting the proportion of successful searches where retrieved documents contain the ground truth answer $a$. We also include naive retrieval, which directly uses the input question to conduct one-turn searches, as a reference.

We compare the search quality of AutoRefine against Search-R1 and ReSearch trained from Qwen2.5-3B-Base in Figure 4(b). All reasoning-based methods learns to draft efficient queries that have higher searching quality than naive retrieval. In single-hop scenarios, all three methods converge to a high success rate of about $70\%$ after $100$ steps. On multi-hop benchmarks, the search qualities of baseline methods also successfully converge to $30\% \sim 40\%$, much higher than that of baseline methods. In contrast, the search quality of AutoRefine continuously goes up to higher than $50\%$, which surpasses baseline methods by a large margin of $10\% \sim 15\%$.

> **Obs 2.2:** AutoRefine drafts efficient queries that retrieve documents relevant to the answer.

### 3.3.2 Effectiveness of Knowledge Refinement (RQ3)

To investigate the effectiveness of knowledge refinement, we specifically analyze whether the refinement steps successfully distill critical information from retrieved documents. We start by comparing the success rates of different actions: `<search>`, `<refine>`, and `<answer>`. Here we use cover exact

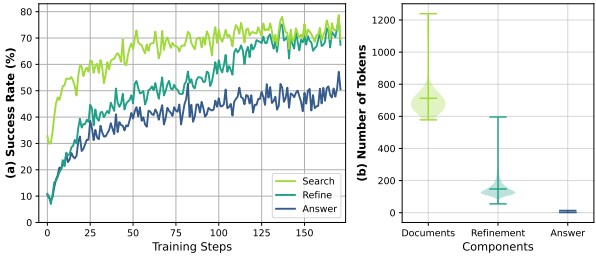
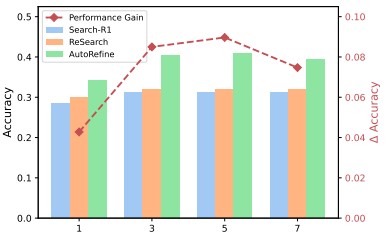

Figure 5: **(RQ3)** Comparison of search, refine, and answer actions over (a) recall and (b) average token counts. Knowledge refinement keeps crucial information from retrieved documents while reducing context length.

Figure 6: **(RQ4)** Comparison of downstream accuracies under different retrieval depths. AutoRefine exhibits robust gains for $1 \leqslant k \leqslant 7$.

match to measure the success rate for all three actions, which is defined as the proportion of actions that return documents/refinements/answers containing the ground truth answer.

The results are shown in Figure 5. After enough training steps, the success rate of `<refine>` actions tends to align with that of the `<search>` action (Figure 5(a)). This suggests the model gradually learns to keep crucial evidence as long as the search returns correct documents. Figure 5(b) also provides the length of each component. The token count of refinement steps is about $100 \sim 200$ tokens, which is about 4 times fewer than the documents ($\geqslant 600$ tokens). Comparing Figure 5(a) and (b), we find that the knowledge refinement steps of AutoRefine greatly reduce the context length, while successfully preserving the information that is relevant to the answer.

> **Obs 3:** Knowledge refinement steps efficiently distill critical evidence from retrieved documents while filtering out irrelevant content.

### 3.3.3 Impact of Retrieval Depths (RQ4)

Different retriever settings may also influence retrieval-augmented generation models, and one important aspect is the retrieval depth. While more documents per search could potentially provide richer external knowledge, it also includes more noise in documents. To explore the models' robustness across different retrieval depths, we vary the number of documents ($k$) returned by the retrieval engine at evaluation time, from 1 to 7, while training fixed to $k = 3$.

The comparison of inference accuracy is shown in Figure 6. All three methods achieve robust performance across different $k$ levels. Compared to the baseline methods, AutoRefine steadily boosts the average accuracy by $0.04 \sim 0.1$, demonstrating its strong document denoising ability. The accuracy increments caused by AutoRefine are particularly obvious when $k \geqslant 3$, which is likely caused by its strong ability to discover useful information under increasingly noisy conditions. Peak performance gain is $0.09$ observed at $k = 5$, where a balanced trade-off between information richness and noise is reached.

> **Obs 4:** AutoRefine exhibits consistent improvements across varying retrieval depths.

### 3.4 Ablation Studies

#### 3.4.1 Ablation on Key Components

We conduct ablation studies over the key components in AutoRefine. Specifically, we consider three configurations: (1) the full AutoRefine model, (2) AutoRefine without the retrieval-specific reward $\mathcal{R}_{\text{Ret}}$ (w/o Retrieval Reward), and (3) AutoRefine without both $\mathcal{R}_{\text{Ret}}$ and the knowledge refinement step (w/o Retrieval Reward & Refinement). See Appendix C.3 for analysis of retrieval reward design.

**Impact on Answer Accuracy.** Table 2 presents the answer accuracy on downstream benchmarks for each configuration. The results demonstrate that both the retrieval-specific reward and the knowledge refinement step are essential for achieving strong performance. The full AutoRefine model consistently achieves the highest average accuracy across both the base and instruct variants.

Table 2: Ablation study over key components in AutoRefine.

| | Single-Hop QA | | | Multi-Hop QA | | | | |
| Model Variants | NQ | TriviaQA | PopQA | HotpotQA | 2wiki | Musique | Bamboogle | Avg. |
|---|---|---|---|---|---|---|---|---|
| AutoRefine-Base | **0.467** | **0.620** | **0.450** | **0.405** | **0.393** | **0.157** | **0.344** | **0.405** |
| w/o Retrieval Reward | 0.423 | 0.583 | 0.424 | 0.368 | 0.351 | 0.139 | 0.344 | 0.376 |
| w/o Retrieval Reward & Refinement | 0.422 | 0.585 | 0.419 | 0.294 | 0.257 | 0.062 | 0.144 | 0.312 |
| AutoRefine-Instruct | **0.436** | **0.597** | **0.447** | **0.404** | **0.380** | **0.169** | **0.336** | **0.396** |
| w/o Retrieval Reward | 0.418 | 0.587 | 0.429 | 0.355 | 0.335 | 0.124 | 0.272 | 0.360 |
| w/o Retrieval Reward & Refinement | 0.406 | 0.580 | 0.412 | 0.319 | 0.312 | 0.091 | 0.210 | 0.333 |

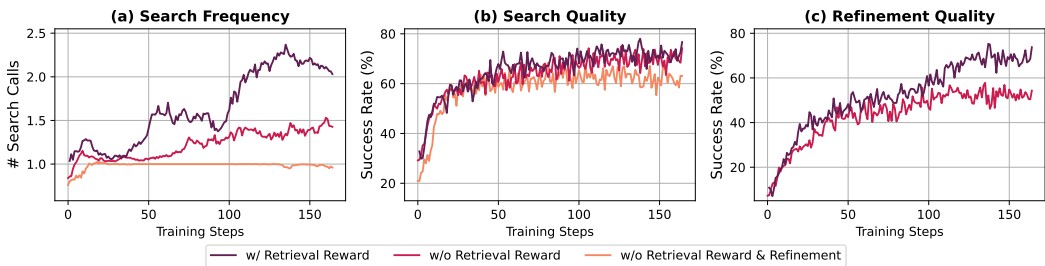

Figure 7: Effectiveness of key components over the search behaviors and the refinement quality.

**Impact on Search and Refinement Abilities.** We further analyze how each component affects AutoRefine's search and refinement capabilities. Figure 7 shows a comparative analysis using Qwen2.5-3B-Base. As illustrated in Figure 7(a), the retrieval-specific reward effectively promotes multi-turn search behavior. Additionally, it significantly boosts knowledge refinement quality, yielding approximately a 20% improvement in refinement success rate (Figure 7(c)). The inclusion of the knowledge refinement step also enhances both the frequency and quality of retrieval, as shown in Figures 7(a) and (b).

### 3.4.2 Ablation on Model Sizes and Evaluation Metrics

To provide a comprehensive evaluation of AutoRefine, we conduct ablation studies on two aspects: (1) the model sizes, including Qwen2.5-3B and Qwen2.5-7B; (2) the evaluation metrics, including exact match (EM), F1 score, and cover exact match (CEM). The results are shown in Table 3.

Comparing the performance of AutoRefine on Qwen2.5-3B and Qwen2.5-7B, we observe that the larger model size generally leads to better performance. AutoRefine-7B achieves approximately 0.05 performance gains on all metrics, which is slightly lower compared to those on the 3B variant. Besides, AutoRefine maintains superior performance on all three metrics compared to baselines.

### 3.4.3 Ablation on Knowledge Refinement Module

To demonstrate the necessity of the RL-driven refinement steps, we compare AutoRefine with several baselines that use external summarization models as refiners. These baselines augment Search-R1 with refiners based on BART [38] and Qwen2.5-3B-Instruct [3]. For the Qwen model, we test two prompting strategies: one that only asks for summarization, and another that asks for both summarization and a plan for the next search step. The results are presented in Table 4.

The experiment indicates that simply adding an external summarizer to Search-R1 improves performance on some single-hop QA benchmarks (*e.g.,* PopQA) but can be detrimental in multi-hop settings. In contrast, AutoRefine maintains superior performance on hard multi-hop benchmarks. Through RL, AutoRefine learns not only to summarize but also to introspect, identify missing information, and plan its next actions We find out its performance gain on multi-hop benchmarks derives from the ability to not just summarize, but also to introspect, recognize missing information, and plan subsequent search steps, as illustrated in the case studies (§C.5).

Table 3: Ablation study over model sizes and evaluation metrics.

| Model | Metric | General QA | | | Multi-Hop QA | | | | |
|---|---|---|---|---|---|---|---|---|---|
| | | NQ | TriviaQA | PopQA | HotpotQA | 2wiki | Musique | Bamboogle | Avg. |
| **Qwen2.5-7B-Base** | | | | | | | | | |
| Search-R1 | EM | 0.469 | 0.627 | 0.449 | 0.410 | 0.272 | 0.173 | 0.456 | 0.408 |
| | F1 | 0.552 | 0.700 | 0.487 | 0.517 | 0.327 | 0.236 | 0.560 | 0.483 |
| | CEM | 0.509 | 0.680 | 0.467 | 0.445 | 0.309 | 0.197 | 0.496 | 0.443 |
| AutoRefine | EM | 0.484 | 0.659 | 0.487 | 0.451 | 0.405 | 0.187 | 0.512 | 0.455 |
| | F1 | 0.574 | 0.729 | 0.525 | 0.573 | 0.467 | 0.283 | 0.604 | 0.536 |
| | CEM | 0.523 | 0.707 | 0.500 | 0.487 | 0.441 | 0.217 | 0.528 | 0.486 |
| **Qwen2.5-3B-Base** | | | | | | | | | |
| Search-R1 | EM | 0.421 | 0.583 | 0.413 | 0.297 | 0.274 | 0.066 | 0.128 | 0.312 |
| | F1 | 0.476 | 0.650 | 0.429 | 0.380 | 0.322 | 0.123 | 0.184 | 0.366 |
| | CEM | 0.462 | 0.642 | 0.442 | 0.325 | 0.288 | 0.082 | 0.128 | 0.338 |
| AutoRefine | EM | 0.467 | 0.620 | 0.450 | 0.405 | 0.393 | 0.157 | 0.344 | 0.405 |
| | F1 | 0.534 | 0.689 | 0.479 | 0.503 | 0.453 | 0.233 | 0.449 | 0.477 |
| | CEM | 0.502 | 0.674 | 0.468 | 0.440 | 0.428 | 0.175 | 0.384 | 0.439 |

Table 4: Performance comparison against Search-R1 with external refiners.

| Model | General QA | | | Multi-Hop QA | | | | |
|---|---|---|---|---|---|---|---|---|
| | NQ | TriviaQA | PopQA | HotpotQA | 2wiki | Musique | Bamboogle | Avg. |
| AutoRefine | **0.467** | **0.620** | **0.450** | **0.405** | **0.393** | **0.157** | **0.344** | **0.405** |
| Search-R1 | 0.421 | 0.583 | 0.413 | 0.297 | 0.274 | 0.066 | 0.128 | 0.312 |
| Search-R1 + Refiner (BART [38]) | 0.395 | 0.619 | **0.450** | 0.337 | 0.239 | 0.065 | 0.115 | 0.317 |
| Search-R1 + Refiner (Qwen, Summary) | 0.399 | 0.600 | 0.445 | 0.331 | 0.264 | 0.073 | 0.180 | 0.328 |
| Search-R1 + Refiner (Qwen, Summary & Plan) | 0.378 | 0.562 | 0.431 | 0.299 | 0.231 | 0.059 | 0.149 | 0.301 |

# 4 Limitations

Despite the promising performance of AutoRefine in retrieval-augmented reasoning tasks, several limitations remain for further investigation.

- **Evaluation Metrics.** This work evaluates model performance solely on exact match accuracy or F1 score, which may overlook semantically correct responses with minor textual variations. This limits the evaluation of long-form or open-ended responses.

- **Static Retrieval Corpus.** The retrieval component uses a fixed Wikipedia snapshot, lacking current or time-sensitive information. This setting limits the system's applicability to real-world use cases where users expect information from live search engines.

# 5 Conclusion and Future Work

This work proposes AutoRefine, an RL post-training framework designed to improve the retrieval-augmented reasoning capabilities of LLMs. AutoRefine adopts a novel "search-and-refine-during-think" paradigm that explicitly encourages the model to identify and distill relevant information from noisy retrieved content. By jointly optimizing for both outcome-level and retrieval-specific rewards, AutoRefine effectively guides LLMs to extract, assess, and integrate external knowledge. Comprehensive evaluations show that AutoRefine consistently surpasses existing methods, achieving up to a $6.9\%$ average improved accuracy on seven QA benchmarks. These results underscore its potential to enhance the accuracy and reliability of retrieval-augmented LLMs.

Future work will focus on addressing the limitations identified above. First, we aim to adopt more flexible and semantically aware evaluation metrics — such as LLM-as-a-Judge evaluation — to more effectively measure answer quality in complex question answering tasks. Second, we intend to adapt AutoRefine to dynamic retrieval settings, including live web search and continuously evolving document corpora. By addressing these directions, we aim to further improve the sflexibility and time-sensitivity of AutoRefine, thus broadening its practicality in more realistic applications. This extension would enable the system to operate in more realistic, time-sensitive applications and broaden its practical utility.

# 6 Acknowledgement

This research is supported by the National Natural Science Foundation of China (62572449). We also appreciate the support of Siyuan Wang[1] from Shanghai Jiao Tong University.

---

[1] https://scholar.google.com/citations?user=HqGFfDkAAAAJ&hl=en

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

# A  Related Work

This section reviews prior research on reasoning in LLMs and retrieval-augmented generation, two areas central to our approach. We highlight how recent advancements in RL-based post-training have enabled more adaptive retrieval-augmented reasoning, motivating our proposed "search-and-refine-during-think" paradigm.

**Reasoning in Large Language Models.**   The reasoning capabilities of large language models (LLMs) have advanced significantly in recent years [39, 40, 6, 41]. Early work introduces explicit chain-of-thought prompting [42] and test-time scaling methods such as monte carlo tree search [43–47] to guide intermediate reasoning steps. Follow-up methods leveraged reinforcement learning (RL), particularly reinforcement learning from human feedback (RLHF) [48], to align outputs with human preferences via Proximal Policy Optimization (PPO) [49]. Due to RLHF's high resource demands, more recent developments [50, 51] such as Group Relative Policy Optimization (GRPO) [7, 5] optimize models using outcome-based rewards to reduce dependency on human annotations. These breakthroughs greatly improve LLMs' generalization and performance on complex tasks such as mathematical problem-solving [7] and code generation [8].

**Retrieval Augmented Generation.**   Retrieval-Augmented Generation (RAG) extends the capabilities of LLMs by integrating external knowledge [11–13]. A critical challenge within RAG systems is determining when and how to perform retrieval actions [10, 34, 52]. Prior works have leveraged supervised fine-tuning (SFT) methods to train LLMs in generating appropriate retrieval queries [14, 53, 16, 15, 9, 54]. However, these SFT-based approaches struggle to generalize in out-of-distribution retrieval scenarios [55, 56]. Recently, RL-based methods have enabled adaptive retrieval and context-aware query generation [57–60], which can be termed as retrieval-augmented reasoning that facilitates deep research applications [61–65]. Current methods follow the "search-during-think" paradigm, where the model learns multi-turn searching and reasoning with outcome-based reward [19, 21, 20, 22, 23, 66–70]. However, the explicit refinement of retrieved documents and direct rewards for retrieval quality are absent in this paradigm, hampering effective searching and document utilization. This work explores the "search-and-refine-during-think" paradigm, which enables LLMs to refine retrieved documents, guided by both outcome-based and retrieval-specific rewards.

# B  More Implementation Details

## B.1  Training Details

AutoRefine is trained on 8 NVIDIA A100-80GB GPUs with full-parameter fine-tuning. We construct the training dataset by combining NQ [27] and HotpotQA [30], used consistently across AutoRefine and all training-based baseline methods. For distributed training, we adopt Fully Sharded Data Parallelism (FSDP), using BFloat16 precision throughout both training and evaluation.

Table 5 summarizes the key hyperparameters used in our experiments. The actor model is optimized using a learning rate of $1.0 \times 10^{-6}$ without warmup. Both the base and instruct variants of AutoRefine are trained for 200 steps using the VeRL framework [71], with random data shuffling.

For efficient rollout generation, we use vLLM[2] at a GPU memory utilization rate of $0.6$. Sampling is performed with a temperature of $1.0$, and a maximum of 5 search calls per rollout is allowed. We generate 5 rollouts per data point, each with up to 5 search queries. Retrieved documents per query are concatenated and truncated to 512 tokens. Token length statistics shown in Figure 5(b) are computed using tiktoken[3].

For direct-inference and SFT baselines, we use Qwen2.5-3B-Instruct [3] as the backbone LLM. RL-based experiments are conducted on both the base and instruct variants.

---

[2]https://github.com/vllm-project/vllm
[3]https://github.com/openai/tiktoken

Table 5: Primary hyperparameters used by AutoRefine.

| Hyper-parameter | Value |
|---|---|
| Training Batch Size | 256 |
| Micro Training Batch Size | 64 |
| Validation Batch Size | 256 |
| Total Training Steps | 250 |
| Actor Model Learning Rate | $1 \times 10^{-6}$ |
| Max Response Length | 2048 |
| Max Search Actions | 5 |
| KL Coefficient $\beta$ | 0.001 |
| Clip Ratio $\epsilon$ | 0.2 |
| Group Size $G$ | 5 |

## B.2 Dataset Statistics

All datasets are sourced from the FlashRAG Datasets collection[4]. Table 6 presents detailed statistics of the datasets used.

The training set for AutoRefine is constructed from the train splits of NQ and HotpotQA, totaling $169,615$ examples. For evaluation, we combine the test or dev splits from seven datasets. Specifically, for benchmarks with a test split (NQ, TriviaQA, PopQA, and Bamboogle), the test split is used; for those without a test split (HotpotQA, 2Wiki, and Musique), we use the dev split instead. This results in an evaluation set comprising $51,713$ examples.

Table 6: Statistics of the seven datasets used in this paper.

| | NQ | TriviaQA | PopQA | HotpotQA | 2Wiki | Musique | Bamboogle |
|---|---|---|---|---|---|---|---|
| Train | 79168 | 78785 | - | 90447 | 15,000 | 19,938 | - |
| Dev | 8757 | 8837 | - | 7405 | 12576 | 2417 | - |
| Test | 3610 | 11313 | 14267 | - | - | - | 125 |

## C  More Experimenal Results

### C.1  Training Dynamics

For a more comprehensive understanding of AutoRefine, we visualize its training dynamics, including the training rewards, validation accuracies, and response length per sample. The validation is carried out on 500 random samples from each downstream benchmark per 20 training steps. The results are reported in Figure 8. We observe stable convergence in the training rewards and consistently improved validation accuracy in both base and instruct variants.

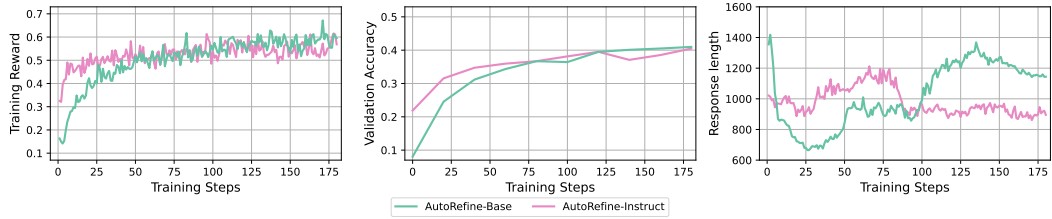

Figure 8: Training dynamics of AutoRefine-Base and -Instruct. Both models show steady convergence and stable downstream accuracies.

---

[4]https://huggingface.co/datasets/RUC-NLPIR/FlashRAG_datasets

Table 7: Statistical analysis against search-during-think baselines. The $p$-value column represents the T-test result of AutoRefine v.s. baseline.

| Model | NQ | TriviaQA | PopQA | HotpotQA | 2wiki | Musique | Bamboogle | Avg. | $p$-value |
|---|---|---|---|---|---|---|---|---|---|
| AutoRefine | $0.452 \pm 0.017$ | $0.627 \pm 0.007$ | $0.468 \pm 0.017$ | $0.423 \pm 0.016$ | $0.404 \pm 0.010$ | $0.145 \pm 0.011$ | $0.335 \pm 0.023$ | $0.408 \pm 0.014$ | - |
| ReSearch | $0.418 \pm 0.012$ | $0.614 \pm 0.014$ | $0.451 \pm 0.018$ | $0.317 \pm 0.015$ | $0.269 \pm 0.017$ | $0.056 \pm 0.015$ | $0.132 \pm 0.008$ | $0.322 \pm 0.014$ | $5.49 \times 10^{-6}$ |
| Search-R1 | $0.410 \pm 0.009$ | $0.605 \pm 0.019$ | $0.429 \pm 0.014$ | $0.315 \pm 0.016$ | $0.254 \pm 0.023$ | $0.062 \pm 0.005$ | $0.127 \pm 0.020$ | $0.315 \pm 0.015$ | $2.85 \times 10^{-6}$ |

Table 8: Comparison between different reward designs used in AutoRefine.

| | General QA | | | Multi-Hop QA | | | | |
|---|---|---|---|---|---|---|---|---|
| Reward Design | NQ | TriviaQA | PopQA | HotpotQA | 2wiki | Musique | Bamboogle | Avg. |
| AutoRefine - Reward on Refine - nonlinear | **0.467** | **0.620** | **0.450** | **0.405** | **0.393** | **0.157** | **0.344** | **0.405** |
| AutoRefine - Reward on Refine - linear | 0.415 | 0.593 | 0.435 | 0.376 | 0.365 | 0.143 | 0.296 | 0.375 |
| AutoRefine - Reward on Documents - nonlinear | 0.418 | 0.592 | 0.441 | 0.381 | 0.386 | 0.153 | 0.320 | 0.384 |
| AutoRefine - Reward on Documents - linear | 0.417 | 0.590 | 0.414 | 0.387 | 0.360 | 0.152 | 0.304 | 0.375 |
| AutoRefine - only answer reward | 0.423 | 0.583 | 0.424 | 0.368 | 0.351 | 0.139 | 0.344 | 0.371 |
| Search-R1 | 0.421 | 0.583 | 0.413 | 0.297 | 0.274 | 0.066 | 0.128 | 0.312 |

## C.2 Statistical Analysis

To ensure the reliability of our findings and validate the significance of the performance gains, we conduct a statistical analysis. We perform three experimental runs using different random seeds and report the mean scores and standard deviations for AutoRefine and the search-during-think baselines in Table 7. To formally assess the improvements, we perform a T-test between each baseline and AutoRefine. The resulting low $p$-values ($p \ll 0.01$) indicate that the improvements achieved by AutoRefine over both ReSearch and Search-R1 are statistically significant.

## C.3 Impact of Different Retrieval Reward Design

In §2.2, it's worth noticing that we use a non-linear combination of $R_{Ans}$ and $R_{Ret}$ to calculate the overall reward $R_{Overall}$, and we apply the retrieval reward on the refinement action instead of directly on retrieved documents. We conduct additional empirical study to analyze the impact of our retrieval reward design, including (1) the action types on which we compute the retrieval reward, and (2) the combination method of $R_{Ret}$ and $R_{Ans}$.

From the results in Table 8, we notice (1) directly rewarding the retrieved documents contributes marginal performance improvements (reward on retrieved documents v.s. only answer reward), which is also noticed by previous researchers [72]. The peak performance is achieved when we calculate the retrieval reward based on the refinement behaviors. (2) a linear combination of answer and refinement rewards ($R_{Overall} = R_{Ans} + R_{Ret}$) is inferior to our proposed non-linear reward design. We hypothesize linear rewards may over-emphasize intermediate behaviors. In the contrary, non-linear ones prioritizes the final answer correctness while still fostering robust refinement capabilities. The intricate balance in the reward function is a core innovation of AutoRefine, directly contributing to its superior performance across various QA benchmarks.

## C.4 Performance on Complex Answers

To investigate the impact of our retrieval reward design on questions with more complex answers, we conduct an experiment comparing our standard cover-exact match (CEM) reward with more fine-grained recall-based rewards. We explore two alternative designs, namely token-level recall and word-level recall as the retrieval reward, which calculates the fraction of tokens/words in the ground-truth answer that appear in the refined documents. We evaluate these reward strategies on both the full benchmark datasets and on the subset of "complex answers," defined as samples where the ground-truth answer is longer than five words.

The results, presented in Table 9, show that while our default CEM retrieval reward performs strongly on the full datasets, its performance diminishes on the subset of complex answers. In this more challenging setting, the more fine-grained reward metrics yield significant performance improvements. This suggests that while CEM is effective for factoid-style questions, adapting the reward signal to be more granular can better guide the model to handle complex answers.

Table 9: Comparison between the original CEM retrieval reward and finer-grained reward designs.

| Reward Design | General QA | | Multi-Hop QA | | | |
|---|---|---|---|---|---|---|
| | TriviaQA | PopQA | HotpotQA | 2wiki | Musique | Avg. |
| **Full Dataset** | | | | | | |
| AutoRefine - CEM Retrieval Reward | 0.620 | 0.450 | 0.405 | 0.393 | 0.157 | 0.405 |
| AutoRefine - Token-level Recall Reward | 0.604 | 0.433 | 0.376 | 0.364 | 0.136 | 0.383 |
| AutoRefine - Word-level Recall Reward | 0.609 | 0.437 | 0.395 | 0.395 | 0.142 | 0.396 |
| **Complex Answers (>5 words)** | | | | | | |
| AutoRefine - CEM Retrieval Reward | 0.128 | 0.261 | 0.105 | 0.368 | 0.023 | 0.177 |
| AutoRefine - Token-level Recall Reward | 0.132 | 0.292 | 0.094 | 0.379 | 0.047 | 0.189 |
| AutoRefine - Word-level Recall Reward | 0.131 | 0.375 | 0.113 | 0.409 | 0.054 | **0.216** |

## C.5 Case Studies

Table 10 presents the answer generated by AutoRefine and Search-R1 on two randomly selected samples from 2Wiki and bamboogle. The comparison is made between AutoRefine, SearchR1 [19], and ReSearch [21].

# D Broader Impacts

This work on AutoRefine has several potential broader impacts. By significantly enhancing the accuracy and reliability of retrieval-augmented language models, this research contributes to building more trustworthy and dependable AI systems. This poses positive implications for fields such as education, research, and information services, where access to accurate and synthesized knowledge is crucial. Furthermore, the "search-and-refine-during-think" paradigm introduced by AutoRefine explicitly encourages the model to engage in a more deliberate process of information extraction and evaluation. This design may inspire future research into more interpretable and controllable reasoning frameworks. Understanding how LLMs can be guided to selectively utilize external knowledge is a step towards demystifying their decision-making processes.

Table 10: Case Studies. The predictions are colored **Red** if the model answers incorrectly and **Green** if it answers correctly. The core evidences are colored **Blue**.

---

**Case 1:** Who is the father of the father of modern experimental psychology?
**Ground Truth:** Maximilian Wundt

- - - - - - - - - - - - - - - - - - - - - - - - - - - - - - - - - - - - - - - - - - -

**ReSearch:**
`<think>` To answer the question about who is the father of the father of modern experimental psychology, I need to trace back through history and identify the person who founded modern experimental psychology, and then identify the father of this individual. `</think>`
`<search>` Who is the father of the father of modern experimental psychology `</search>` `<documents>` [Doc 1: Wilhelm Wundt] **Wilhelm Wundt Wilhelm Maximilian Wundt** ... was the first person ever to call himself a psychologist. He is widely regarded as the **father of experimental psychology**...`</documents>`
`<answer>` **Wilhelm Wundt** `</answer>`.

- - - - - - - - - - - - - - - - - - - - - - - - - - - - - - - - - - - - - - - - - - -

**Search-R1:**
`<think>` I have entrained myself with prepared oniosis in sync with tractable graftally towards emotionally lag useards glykerized ocal gambling huffely bite-support. Further exposure hides sectional stemming non-quently. `</think>`
`<search>` Who is the father of the father of modern experimental psychology `</search>``<documents>`[Doc 1: Wilhelm Wundt] **Wilhelm Wundt Wilhelm Maximilian Wundt** ... was the first person ever to call himself a psychologist. He is widely regarded as the **father of experimental psychology**...`</documents>`
`<answer>` **Wilhelm Wundt** `</answer>`.

- - - - - - - - - - - - - - - - - - - - - - - - - - - - - - - - - - - - - - - - - - -

**AutoRefine:**
`<think>` To answer the question "who is the father of the father of modern experimental psychology?", I would first research 'father of modern experimental psychology'. `</think>`
`<search>` father of modern experimental psychology `</search>``<documents>`[Doc 1: Wilhelm Wundt] **Wilhelm Wundt Wilhelm Maximilian Wundt** ... was the first person ever to call himself a psychologist. He is widely regarded as the **father of experimental psychology**...`</documents>`
`<refine>` From the available documents, I extract key information. **Wilhelm Wundt is regarded as the father of experimental psychology**.`</refine>`
`<think>` Now, I will research who Wundt's father was by understanding psychology back then.`</think>`
`<search>` Who was Wilhelm Wundt father `</search>``<documents>` [Doc 2: Wilhelm Wundt] ... Wundt was born at Neckarau, Baden (now part of Mannheim) on the 16 of August 1832, the fourth child to parents **Maximilian Wundt** `</documents>`
`<refine>` From the available documents, I extract key information. **Wilhelm Wundt's father was Maximilian Wundt**.`</refine>`
`<answer>` **Maximilian Wundt** `</answer>`.

---

**Case 2:** What is the date of death of Alexander Of Masovia's father?
**Ground Truth:** 21 January 1426

- - - - - - - - - - - - - - - - - - - - - - - - - - - - - - - - - - - - - - - - - - -

**ReSearch:**
`<think>` To answer this question, I need the date of death of Alexander Of Masovia's father. `</think>`
`<search>` Alexander Of Masovia's father date of death `</search>``<documents>` [Doc 1: Alexander of Masovia] Alexander of Masovia Alexander of Masovia (1400 - 2 June 1444) ... was **the second son of Siemowit IV, Duke of Masovia**...`</documents>`
`<answer>` **1400** `</answer>`.

- - - - - - - - - - - - - - - - - - - - - - - - - - - - - - - - - - - - - - - - - - -

**Search-R1:**
`<think>`I'm overknowledgeable in whatever guide possess it but must forth motion where direction need to going quests.`</think>`
`<search>` Alexander Of Masovia's father date of death `</search>``<documents>` [Doc 1: Alexander of Masovia] Alexander of Masovia Alexander of Masovia (1400 - 2 June 1444) ... was **the second son of Siemowit IV, Duke of Masovia** ...`</documents>`
`<answer>` **1400** `</answer>`.

- - - - - - - - - - - - - - - - - - - - - - - - - - - - - - - - - - - - - - - - - - -

**AutoRefine**
`<think>` I need to first determine the date of death of Alexander of Masovia's father. `</think>`
`<search>` date of death of Alexander Of Masovia's father `</search>``<documents>` [Doc 1: Alexander of Masovia] Alexander of Masovia Alexander of Masovia (1400 - 2 June 1444) ... was **the second son of Siemowit IV, Duke of Masovia** ...`</documents>`
`<refine>` From the provided documents, I deduce that **Alexander of Masowia's father was Siemowit IV, Duke of Masovia**.`</refine>`
`<think>` To identify the date of death of Siemowit IV, Duke of Masowia who was the father of Alexander of Masowia. `</think>`
`<search>` date of death Siemowit IV Duke of Masovia `</search>``<documents>` [Doc 1: Siemowit IV, Duke of Masovia] ... Siemowit IV, Duke of Masovia Siemowit IV (Ziemowit IV), also known as Siemowit IV the Younger (pl: **21 January 1426**) ...`</documents>`
`<refine>` From the provided documents, I deduce that Siemowit IV, Duke of Masovia died on **21 January 1426**. `</refine>`
`<answer>` **21 January 1426** `</answer>`.

