# OpenReview forum: "Search and Refine During Think: Facilitating Knowledge Refinement for Improved Retrieval-Augmented Reasoning"
_NeurIPS.cc/2025/Conference — NeurIPS 2025 poster_

### Official Review · Reviewer_KVnw · 2025-06-20

**Clarity:** 3
**Significance:** 3
**Originality:** 2
**Rating:** 4
**Confidence:** 4

**Summary:**

This paper proposes a method to enhance retrieval-augmented generation (RAG) by allowing a large language model (LLM) to first think before formulating a search query, and then generate an answer based on the retrieved documents. Building upon prior work such as Search-R1, the authors introduce a novel refine step: after retrieval, the LLM processes the raw documents into a refined form, aiming to prevent misleading information from affecting the final answer. The entire pipeline is trained using reinforcement learning (RL), with a retrieval reward that encourages fetching documents containing the correct answer. Experimental results demonstrate that both the refine step and the retrieval reward contribute to improved performance.

**Questions:**

1. Regarding the retrieval reward: how does the method handle cases where (a) the original documents do not contain the correct answer, or (b) the answer is present in the original documents but lost during refinement? Are these two scenarios treated equivalently during training?
2. In Section 1, the paper claims that SFT struggles to generalize retrieval behaviors to out-of-distribution (OOD) scenarios. Can the authors provide empirical evidence or prior work that supports this claim, especially in the context of retrieval-augmented generation?
3. Is RL truly necessary for learning the refinement step? Have the authors compared against a supervised or prompted alternative (e.g., using gold summaries or instruction-tuned additional refiners) to validate that RL offers a meaningful advantage?

**Ethical Concerns:**

["NO or VERY MINOR ethics concerns only"]

**Final Justification:**

After reading the paper, the rebuttal, and follow-up responses from the authors, I have updated my evaluation as follows:

Resolved Issues:

- Clarity of Contribution: The authors clarified how their refinement module adds to prior work like Search-R1. The response highlighted that RL-based refinement not only summarizes but also supports sub-planning in multi-hop QA, a capability not addressed by external refiners.

- Justification for RL: The authors provided new experiments comparing RL-trained refinement with supervised external summarizers (e.g., BART). These results show that RL offers significant advantages in multi-hop settings, supporting their design choice.

- Title and Framing: The authors acknowledged that the original title was potentially misleading and proposed a clearer alternative. They also committed to improving Figure 1 and better surfacing key comparisons (e.g., Table 3).

Remaining Issues:

- Incomplete Baselines: The comparison with more recent instruction-tuned models (e.g., Qwen2.5, LLaMA3) is still ongoing. While the authors are running these experiments, the lack of results during the rebuttal period leaves the evaluation somewhat incomplete.

- Framing and Presentation: While the authors plan to revise presentation issues in future versions, some of the paper's key motivations (e.g., why RL matters for refinement) were not clearly conveyed in the initial submission.

Summary:

Despite some remaining limitations, I found the authors' response constructive and detailed, and their additional experiments and clarifications meaningfully strengthened the submission. Given the promising empirical results and the novelty of combining RL with refinement in a reasoning-during-retrieval framework, I now view this as a technically sound and useful contribution.

**Limitations:**

Yes

**Quality:**

3

**Strengths And Weaknesses:**

## Strengths

- The proposed method is simple yet effective.
- The paper is well-written and easy to follow, with clear analyses of the method's components.
- The experimental results support the effectiveness of the proposed additions (refinement and reward design).

## Weaknesses

- **Incremental Contribution.** The main idea appears to be an incremental improvement over Search-R1 [1], which already introduced the paradigm of reasoning-before-retrieval trained via RL. The notion that irrelevant context can harm model performance is not new [2], and several previous works have proposed techniques to mitigate this by extracting only relevant information [3]. The contribution of this paper lies primarily in combining document refinement with the Search-R1-style framework and demonstrating its effectiveness, but this alone may be insufficient to warrant acceptance unless further novelty is established.
To better establish the novelty, the authors should articulate the unique challenges or benefits of applying document refinement within a Search-R1-style reasoning-then-retrieval framework, especially in contrast to naive RAG or prior refinement approaches used in more traditional pipelines. Clarifying what makes refinement in this setting particularly necessary or non-trivial (e.g., interaction with the RL objective, refinement timing, or complexity from prior reasoning steps) would significantly strengthen the contribution.
- **Unclear Justification for RL.** In the Introduction, the authors state that “SFT often fails to generalize retrieval behaviors beyond the training distribution.” However, there appears to be no empirical evidence in the paper supporting this claim, nor do the cited references directly address RAG settings. The necessity of RL for both query generation and refinement should be better motivated. Could supervised fine-tuning (SFT) or instruction tuning suffice for learning the refinement step?
- **Misleading Title.** The part of title “Autonomous Retrieval-Augmented Reasoning of LLMs” is somewhat misleading. The core ideas behind autonomous retrieval-augmented reasoning were already developed in Search-R1, while this paper primarily introduces a refinement module and retrieval reward. The current title may give the impression that this paper introduces the full framework from scratch.

---

> ### Author Rebuttal · Authors · 2025-07-30
>
> We sincerely appreciate your thorough review and your positive remarks regarding the conciseness, effectiveness, and delivery of our work. We deeply appreciate your constructive critique regarding our limitations, as it truly helped us strengthen this research.
>
> > W1: The main idea appears to be an incremental improvement over Search-R1 ...
>
> Thank you for your incisive feedback regarding our paper's contribution. You've accurately identified the two core foundations of our work: (1) search-during-reasoning methods and (2) document refinement in RAG. We believe our proposed AutoRefine introduces distinct technical contributions, differentiating it from current approaches in three key aspects: (1) training technique, (2) data requirements, and (3) model behaviors, as summarized in Table 1:
>
> Table 1: Technical Comparison Between Our Work and Previous Methods.
>
> ||training technique||data requirements|model behaviors||
> |-|-|-|-|-|-|
> ||Training Alg|Reward||Reasoning Ability|Refine Behavior|
> |FilCO[1]|SFT|-|Only(Q,A)pairs|-|summarization|
> |IRCoT[2]|-|-|-|√|-|
> |InstructRAG[3]|SFT/ICL|-|(Q,A)+rationale|√|summarization|
> |Search-o1[4]|-|-|-|√|summarization|
> |Self-RAG[5]|SFT|-|(Q,A)+reflect tokens|-|distinguishing|
> |Search-R1[6]|RL|outcome|(Q,A) pairs|√|-|
> |Search-R1-rethink[7]|RL|outcome+retrieval reward|(Q,A) pairs|√|-|
> |AutoRefine|RL|outcome+retrieval reward|(Q,A) pairs|√|summarization+introspection|
>
> As you suggested, we elaborate on the unique challenges and benefits of applying RL-driven refinement within a search-during-reasoning framework.
>
> **[Unique Challenges]** To illustrate the challenges in developing refinement abilities via RL, we conducted **new experiments** providing empirical evidence, as shown in Table 2.
> - Simple paradigm modification is helpful but sub-optimal. Simply changing the reasoning paradigm to "search-and-refine-during-think" does improve overall performance.
> - Retrieval reward on retrieved documents shows limited improvement. Checking whether the answer appears in retrieved documents is also not the ultimate solution, which is noticed by previous research[7]. We hypothesize this is because rewards on documents may shortcut the model to draft more search queries for broader knowledge coverage, rather than more accurate queries and more efficient refinement.
> - Linear reward may over-emphasize intermediate behaviors. An intuitive way is to add $R_{ref}$ with $R_{ans}$ linearly as $R_{overall}=R_{ans}+R_{ref}$. However, we believe that the extra abilities (format-following, refinement) should be foundational yet incidental, as the primary aim is to achieve a correct answer. As a result, non-linear retrieval rewards surpasses linear rewards on both docs/refines.
>
> Table 2: Comparison Between Different Rewards Used in AutoRefine.
>
> ||General QA|||Multi-Hop QA|||||
> |-|-|-|-|-|-|-|-|-|
> ||NQ|TriviaQA|PopQA|HotpotQA|2wiki|Musique|Bamboogle|Avg.|
> |AutoRefine-Reward at Refine-nonlinear|0.467|0.620|0.450|0.405|0.393|0.157|0.344|0.405|
> |  -Reward at Refine-linear|0.415|0.593|0.435|0.376|0.365|0.143|0.296|0.375|
> |  -Reward at Documents-nonlinear|0.418|0.592|0.441|0.381|0.386|0.153|0.320|0.384|
> |  -Reward at Documents-linear|0.417|0.590|0.414|0.387|0.360|0.152|0.304|0.375|
> |  -only answer reward|0.423|0.583|0.424|0.368|0.351|0.139|0.344|0.371|
> |Search-R1[6]|0.421|0.583|0.413|0.297|0.274|0.066|0.128|0.312|
>
> **[Unique Benefits]**
> - No annotation requirements. AutoRefine leverages the intrinsic knowledge refinement ability of LLMs without preliminary search trajectory annotation.
> - Complexity in search-during-think reasoning. We conducted **additional experiments** comparing AutoRefine with using an external refiner model with Search-R1, as shown in Table 3 addressing your Q3. While external refiner achieve good performance on single-hop QA, it causes performance drops on multi-hop ones. Besides summarization, AutoRefine learns to recognize missing information and plan subsequent search steps through RL (Table 5 in our paper). This ability contributes to the superior multi-hop performance.
>
> > W2: Could supervised fine-tuning (SFT) or instruction tuning suffice for learning the refinement step?
>
> We appreciate the reviewer's astute observation regarding the unclear justification in our introduction. We agree that our initial statement, claiming "SFT often fails to generalize retrieval behaviors beyond the training distribution", is **unsubstantiated and overly opinionated** without empirical evidence.
>
> Upon your suggestion, we propose the following revisions to clarify the necessity of RL for training retrieval-augmented LLMs:
> - Revised Justification for RL. We will modify the statement to highlight the practical challenges of SFT. Specifically, below is our revised statement:
>
> "*While SFT can be effective for training large models for search, it sometimes necessitates the construction of high-quality search paths, which incurs additional effort and resource overheads. To address this, recent studies draw inspiration from RL-based post-training and explore RL for retrieval-augmented reasoning, achieving excellent results by only evaluating final answer correctness without the need for pre-collected reasoning paths.*"
> - Addition of Contextual Citations. We will explicitly acknowledge the effectiveness of SFT by adding the following citations [4,8,9,10].
>
> Thank you once again for identifying this crucial flaw in our introduction. We will thoroughly review the remainder of our paper to identify and rectify any similar issues, ensuring our claims are rigorously supported and well-justified.
> We believe these revisions will significantly solidify our work and enhance its contribution to the research community.
>
>
> > W3: The part of title “Autonomous Retrieval-Augmented Reasoning of LLMs” is somewhat misleading.
>
> We sincerely thank the reviewer for their insightful feedback regarding our original title. We agree that it (1) did not fully capture our core innovations, the refinement module and retrieval reward, and (2) could indeed imply a broader scope than intended.
>
> Acting on this valuable suggestion, we've decided to **update our paper's title** to:
>
> *Search and Refine During Think: Facilitating On-the-Fly Knowledge Distillation for Improved Retrieval-Augmented Reasoning*
>
> We believe this new title more accurately reflects the essence of our work by highlighting its key focus (retrieval-augmented reasoning), key novelty (on-the-fly knowledge distillation methodology), and key results (improved performance).
>
> > Q1: Are these two scenarios treated equivalently during training?
>
> Yes. In our method, the retrieval reward specifically reflects the correctness of the refinement process itself.
> We believe that truly effective search-augmented reasoning requires not only retrieving correct information, but also effective use of the information.
>
> As shown in Figure 5(a) in our paper, the search success rate begins relatively high compared to refine and answer, and ends up nearly 80%. From this phenomenon, we hypothesize that **search behaviors are relatively easy to learn**. Thus, we primarily shape the refine behaviors during RL training.
>
> > Q2: Can the authors provide empirical evidence or prior work that supports this claim, especially in the context of retrieval-augmented generation?
>
> Please see our response to W2.
>
> > Q3: Is RL truly necessary for learning the refinement step?
>
> Thank you for raising the excellent point regarding the necessity of RL for the refinement step and the comparison against more direct, supervised approaches. Upon your advice, we conduct **additional experiments** under the intuitive setting: using a fine-tuned external refiner [11] to summarize retrieved documents in Search-R1, and the results are presented in the table below:
>
> Table 3: Comparison Between SFT-based Refiner and RL-based Refinement.
>
> ||GeneralQA|||Multi-HopQA|||||
> |-|-|-|-|-|-|-|-|-|
> ||NQ|TriviaQA|PopQA|HotpotQA|2wiki|Musique|Bamboogle|Avg.|
> |AutoRefine|0.467|0.620|0.450|0.405|0.393|0.157|0.344|0.405|
> |Search-R1|0.421|0.583|0.413|0.297|0.274|0.066|0.128|0.312|
> |Search-R1+external refiner|0.395|0.619|0.450|0.337|0.239|0.065|0.115|0.317|
> |ReSearch|0.427|0.597|0.430|0.305|0.272|0.074|0.128|0.319|
> |ReSearch+external refiner|0.419|0.614|0.445|0.334|0.248|0.069|0.114|0.321|
>
> From these results, we observed that external refiners improves performance on several general QA benchmarks, but leads to a performance decline on multi-hop QA ones.
>
> **[Role of RL]** These observations lead us to the following explanation regarding the distinct roles of external refiners and our RL-trained refinement module:
> - **Single-Hop Scenarios:** In single-hop QA, refinement learned via RL appear to function similarly to external refiners.
> - **Multi-Hop Scenarios (RL's Advantage):** Under multi-hop settings, the LLM learns to introspect missing information (Table 5 in the paper). This phenomenon explains why RL-driven refinement outperforms simply applying external refiners in multi-hop QA.
>
> [1] Learning to filter context for retrieval-augmented generation
>
> [2] Interleaving Retrieval with Chain-of-Thought Reasoning for Knowledge-Intensive Multi-Step Questions
>
> [3] InstructRAG: Instructing Retrieval-Augmented Generation via Self-Synthesized Rationales
>
> [4] Self-RAG: Learning to Retrieve, Generate, and Critique through Self-Reflection
>
> [5] Search-o1: Agentic search-enhanced large reasoning models
>
> [6] Search-r1: Training llms to reason and leverage search engines with reinforcement learning
>
> [7] An Empirical Study on Reinforcement Learning for Reasoning-Search Interleaved LLM Agents
>
> [8] InstructRAG: Instructing Retrieval-Augmented Generation via Self-Synthesized Rationales
>
> [9] Multi-step Retriever-Reader Interaction for Scalable Open-domain Question Answering
>
> [10] Multi-Hop Paragraph Retrieval for Open-Domain Question Answering
>
> [11] BART: Denoising Sequence-to-Sequence Pre-training for Natural Language Generation, Translation, and Comprehension

---

> > ### Comment · Reviewer_KVnw · 2025-08-04
> >
> > Thank you for the detailed rebuttal. I have some follow-up thoughts and clarifications based on your response.
> >
> > 1. **W1:**
> >
> >     After reading the response, I now better appreciate the contribution of this work. The paper presents a clear motivation for refining documents during reasoning, and the RL-based framework is well-aligned with this goal. I now view the significance of the work more positively.
> >
> > 2. **W2/Q3:**
> >
> >     I would like to ask for clarification on the external summarizer used in Table 3. You mention using BART by reference, but I’m curious **why a more recent instruction-tuned model (e.g., Qwen2.5, Llama3) was not used instead**. Given that newer models often perform better with proper instructions, such a comparison could provide a stronger baseline.
> >
> > 3. **Q3 (continued):**
> >
> >     I understand now that the key benefit of RL refinement is its ability to support sub-planning in multi-hop settings without external modules, which is more than summarization. While I agree this is a valuable capability, **it would be helpful if the main paper presented Table 3 more prominently** to help practitioners decide between RL-based and external refinement approaches based on their needs.
> >
> > 4. **Figure 1 (main paper):**
> >
> >     The motivation for refinement via RL, especially its advantage in multi-hop QA, is compelling. However, I feel that this motivation is not clearly conveyed in Figure 1. **Strengthening the figure to better illustrate the refinement challenge** could improve clarity.
> >
> > 5. **Revised Title**:
> >
> >     I was a bit confused by the phrase "on-the-fly knowledge distillation" in the revised title. Typically, knowledge distillation refers to transferring capabilities from a teacher model to a student. Since your method does not involve a teacher-student setup, **could you clarify what exactly is being distilled, and in what sense it is "on-the-fly"?**
> >
> >
> > Overall, this rebuttal clarified several points and raised interesting directions for thinking about how to improve RL-based RAG system training. I'm now considering adjusting my score to a weak accept.

---

> > > ### Author Response · Authors · 2025-08-05
> > > **Reply to further questions by Reviewer KVnw**
> > >
> > > Dear Reviewer KVnw:
> > >
> > > We sincerely thank you for your thoughtful reviews and valuable suggestions, which have been instrumental in improving our work. We are also glad that our previous response successfully addressed some of your concerns.
> > >
> > > Regarding your remaining questions, we are taking the following actions:
> > >
> > > > W2/Q3: why a more recent instruction-tuned model (e.g., Qwen2.5, Llama3) was not used instead?
> > > - We did not include experiments with state-of-the-art LLMs, such as Qwen2.5 or Llama3, due to some technical difficulties that we could not resolve during the limited rebuttal period. Following your suggestion, we are currently running experiments using a Qwen2.5-3B Instruct model, which is prompted under two different settings: (1) provide a summarization of retrieved documents or (2) provide a summarization of retrieved documents and next-step sub-planning. We will update the results as soon as they are available.
> > >
> > > > Q3 (continued): It would be helpful if the main paper presented Table 3 more prominently to help practitioners decide between RL-based and external refinement approaches based on their needs.
> > >
> > > > Figure 1 (main paper): Strengthening the figure to better illustrate the refinement challenge could improve clarity.
> > >
> > > - Your feedback has helped us realize that the original presentation (including Figure 1, the introduction, and the missing comparison with an external refiner) did not fully capture the unique benefits of our method. We will revise the writing and visualizations in future versions of the manuscript to address this.
> > >
> > > > Revised Title: Could you clarify what exactly is being distilled, and in what sense it is "on-the-fly"?
> > > - We use the term "on-the-fly" to convey that our refinement process is executed by the LLM itself during the reasoning phase, rather than by an external component. We agree that the terms "on-the-fly" and "distillation" may cause unnecessary confusion. Therefore, removing them will make our title more direct and better reflect our key insight. We propose the following revised title:
> > >   - *Search and Refine During Think: Facilitating Knowledge Refinement for Improved Retrieval-Augmented Reasoning*
> > >
> > > Thank you again for all the efforts you've made to improve our manuscript.
> > >
> > > Best regards,
> > >
> > > The Authors

---

> > > > ### Comment · Reviewer_KVnw · 2025-08-05
> > > >
> > > > Thank you for the response. I appreciate the authors' efforts to address the remaining concerns, including the ongoing suggested Qwen2.5-3B Instruct experiments, plan to highlight Table 3 more prominently, strengthen Figure 1, and clarify the title. I hope these revisions will be reflected in future versions of the paper.
> > > >
> > > > I have accordingly raised my score to 4.

---

> > > > > ### Author Response · Authors · 2025-08-06
> > > > > **Reply to Comment by Reviewer KVnw**
> > > > >
> > > > > Dear Reviewer KVnw,
> > > > >
> > > > > Thanks for your feedback. We have finished the experiments using Qwen2.5-3B-Instruct as the external refiner. We now have the new experimental results, where we prompt Qwen2.5-3B-Instruct in two settings:
> > > > > 1. Ask the model to summarize the documents
> > > > > 2. Ask the model to summarize the documents, and make plans for the next search step
> > > > >
> > > > > The results are shown in the table below.
> > > > >
> > > > > Table: Performance Comparison with external refiners
> > > > >
> > > > > |                                                                               | General QA |          |       | Multi-Hop QA |       |         |           |       |
> > > > > | ----------------------------------------------------------------------------- | ---------- | -------- | ----- | ------------ | ----- | ------- | --------- | ----- |
> > > > > |                                                                               | NQ         | TriviaQA | PopQA | HotpotQA     | 2wiki | Musique | Bamboogle | Avg.  |
> > > > > | AutoRefine                                                                    | **0.467**      | **0.620**    | **0.450** | **0.405**        | **0.393** | **0.157**   | **0.344**     | **0.405** |
> > > > > | FaviComp                                                                      | 0.302      | 0.502    | 0.410 | 0.240        | 0.220 | 0.054   | 0.232     | 0.280 |
> > > > > | Search-R1 + external refiner (BART)                                           | 0.395      | 0.619    | **0.450** | 0.337        | 0.239 | 0.065   | 0.115     | 0.317 |
> > > > > | Search-R1 + external refiner (Qwen2.5-3B-Instruct, summary)                   | 0.399      | 0.600    | 0.445 | 0.331        | 0.264 | 0.073   | 0.180     | 0.328 |
> > > > > | Search-R1 + external refiner (Qwen2.5-3B-Instruct, summary & plan) | 0.378      | 0.562    | 0.431 | 0.299        | 0.231 | 0.059   | 0.149     | 0.301 |
> > > > >
> > > > > The above results indicate that (1) the more up-to-date Qwen2.5 model is indeed better than BART at summarization (0.011 overall improvement), which meets your expectations, and (2) AutoRefine's RL-driven refinement behaviors are more effective than external refiners.
> > > > >
> > > > > We hope the updated results have addressed your further concerns. Thank you again for your positive opinions on our work.
> > > > >
> > > > > Best regards,
> > > > >
> > > > > The Authors

---

> ### Author Response · Authors · 2025-08-03
> **Looking forward to your reply**
>
> Dear Reviewer KVnw,
>
> Thank you again for your constructive suggestions. We have carefully addressed your concerns by:
> - systematically comparing previous methods and ours
> - elaborating the unique challenges and benefits of this work
> - modifying the misleading statement & title
> - demonstrating the effectiveness of RL-driven refinement with new experiments
>
> We look forward to engaging in further discussion with you and would greatly appreciate your positive feedback on our rebuttal.
>
> Best regards,
>
> The Authors

---

### Official Review · Reviewer_UA8w · 2025-07-03

**Clarity:** 4
**Significance:** 2
**Originality:** 3
**Rating:** 5
**Confidence:** 2

**Summary:**

The paper studies the problem of training an LLM to issue search queries to enable RAG. Based on the observation that prior works have mostly focused on eliciting search skills in the LM via SFT, the proposed method uses RL to train a reader model. While the proposed paper is not the first to use RL for training a reader, they make two key contributions:
1. the proposed method, AutoRefine, flexibly summarizes (“refines”) retrieved documents in addition to retrieving, akin to https://arxiv.org/abs/2311.08377
2. the proposed method uses process rewards for retrieval

On a variety of question answering benchmarks, the proposed method is shown to be very promising.

**Questions:**

- The retriever reward uses a binary reward based on term matching of the answer with the ground truth. This seems to effectively be training the model to prefer exact match queries, which may not be globally optimal. Is this a legitimate concern?

**Ethical Concerns:**

["NO or VERY MINOR ethics concerns only"]

**Final Justification:**

Final score: 5 (accept).

After reviewing the author's responses and other reviewers' comments, I maintain my score of a 5. I agree with reviewer jRtt that this paper is largely incremental. It is conceptually just combining two existing paradigms: filtering context for RAG [1] and improving agentic search via RL [2]. I agree with reviewer jRtt that the resemblance to prior work is understated, and I believe it is concerning that [1] was not cited in this paper.

[1] Wang, Zhiruo, et al. "Learning to filter context for retrieval-augmented generation." arXiv preprint arXiv:2311.08377 (2023).
[2] Jin, Bowen, et al. "Search-r1: Training llms to reason and leverage search engines with reinforcement learning." arXiv preprint arXiv:2503.09516 (2025).

**However**, unlike reviewer jRtt, I don't believe that an incremental contribution is inappropriate for NeurIPS. I think this paper demonstrates that RL training is useful for evidence refinement/context filtering, and I think the methods used to achieve that (including process rewards) are quite instructive.


TL;DR: incremental contribution, well-executed, likely to be useful to the community

**Limitations:**

Yes.

**Quality:**

4

**Strengths And Weaknesses:**

Strengths:
- The experiments in this paper are very well-chosen and give a complete picture of the proposed method.
- The empirical support for the efficacy of this method is strong for multi-hop QA.

Weaknesses:
- It seems that the two methodological contributions of this paper ("Refine" and "Process Rewards for Search") each help on different benchmarks. This suggests that potentially these two methods are actually more targeted. I was hoping to see more discussion of this.
- The retriever reward uses a binary reward based on term matching of the answer with the ground truth. This seems to effectively be training the model to prefer exact match queries, which may not be globally optimal. I'd like to see more discussion of the possible risks of this approach.
- As mentioned by the authors in the limitation section, the proposed method is only demonstrated with 3B models, which are a bit smaller than standard for readers in a RAG system these days (8B has become the norm in academic work). It's likely that 7B or 14B models will benefit less from carefully-designed inductive biases such as the ones proposed in this paper. In any case, this does not invalidate the value of the paper.

---

> ### Author Rebuttal · Authors · 2025-07-30
>
> We're grateful for your insightful review and appreciate your recognition of our work's comprehensive evaluation and outstanding efficacy. We have addressed your major concerns in detail below.
>
> > **Weakness 1: More Explanation over Methodological Contributions.** It seems that the two methodological contributions of this paper ("Refine" and "Process Rewards for Search") each help on different benchmarks. This suggests that potentially these two methods are actually more targeted. I was hoping to see more discussion of this.
>
> **Response:**
> We appreciate your insightful suggestion regarding the distinct impacts of our methodological contributions.
>
> We agree that our two main contributions, "Knowledge Refinement Step" and "Processing Rewards for Search" (retrieval reward), exhibit different performance characteristics across benchmarks. To better answer the question, we **extend our ablation study** with a new control group that combines Search-R1 with an external fine-tuned summarization model (`facebook/bart-large-cnn` [1]), as shown in Table 1. In the new setting, the external model summarizes retrieved documents and appends the summarization at the end of the retrieval.
>
> **Table 1:** Extended Ablation Study.
>
> |                                 | General QA |          |       | Multi-Hop QA |       |         |           |       |
> | ------------------------------- | ---------- | -------- | ----- | ------------ | ----- | ------- | --------- | ----- |
> |                                 | NQ         | TriviaQA | PopQA | HotpotQA     | 2wiki | Musique | Bamboogle | Avg.  |
> | AutoRefine                      | 0.467      | 0.620    | 0.450 | 0.405        | 0.393 | 0.157   | 0.344     | 0.405 |
> | - retrieval reward              | 0.423      | 0.583    | 0.424 | 0.368        | 0.351 | 0.139   | 0.344     | 0.376 |
> | - retrieval reward & refinement | 0.422      | 0.585    | 0.419 | 0.294        | 0.257 | 0.062   | 0.144     | 0.312 |
> | Search-R1 + external summarization | 0.395      | 0.619    | 0.450 | 0.337        | 0.239 | 0.065   | 0.115     | 0.317 |
>
> We observe the following:
> - **External Document Summarization Boosts Single-hop Performance.** Search-R1's single-hop QA performance (TriviaQA, PopQA) is approximately the same as AutoRefine when equipped with an external summarization module. This indicates that mere summarization is sufficient in such simple scenarios. On the contrary, the multi-hop QA performance slightly drops.
> - **Refinement Step Boosts Multi-hop Performance.** Incorporating refinement steps in the reasoning paths significantly improves the model's performance on multi-hop benchmarks, while keeping the single-hop performance unchanged. What promotes such solid multi-hop performance? Taking a closer look at the case studies (Table 5 in our paper), we find that the model evolves the introspection behavior - it spontaneously reflects missing information in the retrieved documents and
> - **Retrieval Reward Yields Comprehensive Improvements.** However, merely adding a refinement step does not fully develop refinement behavior. When trained with the retrieval reward, we observe a comprehensive improvement across both single- and multi-hop benchmarks. Notably, the model exhibits similar performance on single-hop benchmarks as vanilla Search-R1 with an external summarization module, which suggests that the performance gain stems from the enhanced summarization capability of the LLM.
>
> We hope our fine-grained explanation with additional experiments can address your concerns.
>
> > **Weakness 2: Possible Risk of EM Retrieval Reward.** The retriever reward uses a binary reward based on term matching of the answer with the ground truth. This seems to effectively be training the model to prefer exact match queries, which may not be globally optimal. I'd like to see more discussion of the possible risks of this approach.
>
> **Response:**
>
> Thanks for pointing out the potential risk in the strict cover EM retrieval reward. After careful consideration, we hypothesize that the strict reward may harm the models' performance when the answer is complex.
>
> We add **new experiments**, comparing our current retrieval reward against two more flexible designs:
> 1. Token-level recall: Calculates how many *tokens* in the ground-truth answer appear in the refinement,
> 2. Word-level recall: Calculates how many *words* in the ground-truth answer appear in the refinement;
>
> under different benchmark settings:
> - Full dataset: Test on all questions in the benchmark.
> - Complex answers: Test on questions with an answer >5 words.
>
> The model is evaluated with EM, and the results are summarized in Table 2.
>
> **Table 2:** Comparison between original and finer-grained retrieval reward design.
>
> |                                                  | **General QA** |           |           | **Multi-Hop QA** |           |           |
> | ------------------------------------------------ | -------------- | --------- | --------- | ---------------- | --------- | --------- |
> |                                                  | TriviaQA       | PopQA     | HotpotQA  | 2wiki            | Musique   | Avg.      |
> | **Full Dataset**                                 |                |           |           |                  |           |           |
> | AutoRefine - CEM Retrieval Reward                | **0.620**      | **0.450** | **0.405** | 0.393            | **0.157** | **0.405** |
> | AutoRefine - Token-level Recall Retrieval Reward | 0.604          | 0.433     | 0.376     | 0.364            | 0.136     | 0.383     |
> | AutoRefine - Word-level Recall Retrieval Reward  | 0.609          | 0.437     | 0.395     | **0.395**        | 0.142     | 0.396     |
> | **Complex Answers  (>5 words)**                  |                |           |           |                  |           |           |
> | AutoRefine - CEM Retrieval Reward                | 0.128          | 0.261     | 0.105     | 0.368            | 0.023     | 0.177     |
> | AutoRefine - Token-level Recall Retrieval Reward | **0.132**      | 0.292     | 0.094     | 0.379            | 0.047     | 0.189     |
> | AutoRefine - Word-level Recall Retrieval Reward  | 0.131          | **0.375** | **0.113** | **0.409**        | **0.054** | **0.216** |
>
> The results confirm that though the cover EM reward works well on most QA cases, it could **bring potential risks** such as performance degradation **when the target answer is complex**.
>
> > **Weakness 3: Performance at Different Model Scales.** As mentioned by the authors in the limitation section, the proposed method is only demonstrated with 3B models, which are a bit smaller than standard for readers in a RAG system these days (8B has become the norm in academic work). It's likely that 7B or 14B models will benefit less from carefully-designed inductive biases such as the ones proposed in this paper. In any case, this does not invalidate the value of the paper.
>
> **Response:**
> We appreciate you raising the point about model scale and its potential impact on our method's effectiveness. We agree that larger models (e.g., 7B or 14B) might inherently benefit less from explicitly designed inductive biases.
>
> We conduct **additional experiments** to demonstrate AutoRefine's effectiveness at the 7B level, and the results are shown in Table 3. The baseline result (Search-R1) is reproduced using their officially released checkpoint.
>
> **Table 3:** Performance Comparison at Different Model Scales
>
> |                 | General QA |          |       | Multi-Hop QA |       |         |           |       |
> | --------------- | ---------- | -------- | ----- | ------------ | ----- | ------- | --------- | ----- |
> |                 | NQ         | TriviaQA | PopQA | HotpotQA     | 2wiki | Musique | Bamboogle | Avg.  |
> | **Qwen2.5-7B-Base** |            |          |       |              |       |         |           |       |
> | Search-R1       | 0.469      | 0.627    | 0.449 | 0.410        | 0.272 | 0.173   | 0.456     | 0.408 |
> | AutoRefine      | 0.484      | 0.659    | 0.487 | 0.451        | 0.405 | 0.187   | 0.512     | 0.455 |
> | **Qwen2.5-3B-Base** |            |          |       |              |       |         |           |       |
> | Search-R1       | 0.421      | 0.583    | 0.413 | 0.297        | 0.274 | 0.066   | 0.128     | 0.312 |
> | AutoRefine      | 0.467      | 0.620    | 0.450 | 0.405        | 0.393 | 0.157   | 0.344     | 0.405 |
>
> From the Table, we observe that AutoRefine still yields positive results even at the 7B parameter level, demonstrating an overall performance improvement of approximately 5%.
>
> This gain is indeed slightly less than the significant 9% improvement observed with 3B models, which aligns with your hypothesis that the benefits from explicit refinement steps might relatively diminish as model scale increases.
>
> > **Question 1: Potential Risk in Strict Retrieval Reward.** The retriever reward uses a binary reward based on term matching of the answer with the ground truth. This seems to effectively be training the model to prefer exact match queries, which may not be globally optimal. Is this a legitimate concern?
> - Please see our response to your **Weakness 2**.
>
> [1] Lewis, Mike, et al. "BART: Denoising Sequence-to-Sequence Pre-training for Natural Language Generation, Translation, and Comprehension." _Proceedings of the 58th Annual Meeting of the Association for Computational Linguistics_. 2020.

---

> ### Author Response · Authors · 2025-08-03
> **Looking forward to your reply**
>
> Dear Reviewer UA8w,
>
> Thank you again for your constructive comments and positive opinions. We have carefully addressed your concerns by:
> - making more explanation of methodological contributions with new empirical evidence
> - exploring new reward designs to see the practicability of binary retrieval reward
> - successfully expand our method to a larger model scale
>
> We look forward to further discussion with you and would greatly appreciate your positive feedback on our rebuttal.
>
> Best regards,
>
> The Authors

---

> > ### Comment · Reviewer_UA8w · 2025-08-05
> > **Thank you for your detailed rebuttal!**
> >
> > The study provided of manual summarization during refinement is useful. Thank you for doing that! This does suggest that using a frozen summarizer module to do context refinement is sufficient for simple factoid retrieval tasks but insufficient for complex tasks -- this is an interesting and intuitive finding. However, my primary concern here is whether BART-large is a fair comparison with the proposed refinement module. Given the advances in scale and data since BART, simply prompting Qwen2.5-3B-Instruct to do retrieval would be a more instructive comparison.
> >
> > Regarding your follow-up experiments about "Possible Risk of EM Retrieval Reward", I think it's interesting that the strict-cover exact-match reward is clearly much more effective than token-level or word-level recall metrics, but I actually think this does not address my original concern. These are all forms of lexical matching, and none of these would prevent the model from, in theory, learning to produce BM25-style retrieval behavior. **That being said, upon second examination, since the reward used here is lexical matching _with the ground truth answer_ from NQ/HotpotQA**, the use of exact-match rewards from the retriever seems like a reasonable enough thing to do. I no longer feel this is a cause for concern.
> >
> > Finally, thank you for your experiments with scaling up the reader! This does align with my original intuitions that, as the base model becomes bigger, the number of errors that clever post-training can prevent will necessarily go down. Nonetheless, 5\% aggregate improvement over Search-R1-7B is still a good improvement.
> >
> > In summary, I maintain my positive evaluation of 5. I ultimately agree with reviewer jRtt that the proposed work is fairly incremental, but I think that's absolutely ok -- this work would still be useful to the NeurIPS community.

---

> > > ### Author Response · Authors · 2025-08-06
> > > **Reply to Comment by Reviewer UA8w**
> > >
> > > Dear Reviewer UA8w,
> > >
> > > Thank you for your careful re-evaluation of our work and such detailed feedback. We are grateful for your positive assessment and for the time you took to reconsider your initial concerns, particularly regarding the retrieval reward design.
> > >
> > > Regarding your remaining questions about the external refiner choice:
> > > > However, my primary concern here is whether BART-large is a fair comparison with the proposed refinement module. Given the advances in scale and data since BART, simply prompting Qwen2.5-3B-Instruct to do retrieval would be a more instructive comparison.
> > >
> > > We did not include experiments using Qwen2.5 as the external refiner due to some technical difficulties that we were unable to resolve within the limited rebuttal period. We now have the new experimental results, where we prompt Qwen2.5-3B-Instruct in two settings:
> > > 1. Ask the model to summarize the documents
> > > 2. Ask the model to summarize the documents, and make plans for the next search step
> > >
> > > And the results are shown in the table below.
> > >
> > > **Table:** Performance Comparison with external refiners
> > >
> > > |                                                                               | General QA |          |       | Multi-Hop QA |       |         |           |       |
> > > | ----------------------------------------------------------------------------- | ---------- | -------- | ----- | ------------ | ----- | ------- | --------- | ----- |
> > > |                                                                               | NQ         | TriviaQA | PopQA | HotpotQA     | 2wiki | Musique | Bamboogle | Avg.  |
> > > | AutoRefine                                                                    | **0.467**      | **0.620**    | **0.450** | **0.405**        | **0.393** | **0.157**   | **0.344**     | **0.405** |
> > > | FaviComp                                                                      | 0.302      | 0.502    | 0.410 | 0.240        | 0.220 | 0.054   | 0.232     | 0.280 |
> > > | Search-R1 + external refiner (BART)                                           | 0.395      | 0.619    | **0.450** | 0.337        | 0.239 | 0.065   | 0.115     | 0.317 |
> > > | Search-R1 + external refiner (Qwen2.5-3B-Instruct, summary)                   | 0.399      | 0.600    | 0.445 | 0.331        | 0.264 | 0.073   | 0.180     | 0.328 |
> > > | Search-R1 + external refiner (Qwen2.5-3B-Instruct, summary & plan) | 0.378      | 0.562    | 0.431 | 0.299        | 0.231 | 0.059   | 0.149     | 0.301 |
> > >
> > > The above results indicate that (1) the newer Qwen2.5 model is indeed better than BART at summarization (0.011 aggregated improvement), which meets your expectations; (2) AutoRefine's RL-driven refinement behaviors are more effective than external refiners.
> > >
> > > We hope the above results have addressed your further concerns. Thank you again for your recognition of our paper.
> > >
> > > Best regards,
> > >
> > > The Authors

---

### Official Review · Reviewer_9S2E · 2025-07-03

**Clarity:** 2
**Significance:** 2
**Originality:** 2
**Rating:** 4
**Confidence:** 5

**Summary:**

The paper presents a method for enhancing reinforcement learning (RL) training in retrieval-augmented question answering through evidence compression. It introduces soft rewards designed to promote faithful answer generation based on retrieved context. The approach is evaluated on diverse Wikipedia-based short-form single- and multi-hop open-ended QA tasks, demonstrating performance gains compared to existing baselines.

**Questions:**

1. Hyperparameter justification: The paper lacks explicit rationale for key hyperparameters (e.g., choice of E5 as retriever, 3 documents fetched during training). If informed by prior work, this should be clearly stated to demonstrate scientific grounding.

2. Reward function ablation: The reward function’s design (e.g., 0.1 as partial reward, no partial reward for correct final answers) is not analyzed. This omission limits understanding of the method’s effectiveness and trade-offs.

3.  Training configuration ambiguity: The choice of maximum search calls (5) and whether training settings align with Search-R1 are unclear. Section B.1 and Table 3 do not provide sufficient details to verify reproducibility or consistency with prior work.

4. Data split specification: The dev split of Hotpot-QA (distractor vs. fullwiki) is not explicitly stated, creating ambiguity about the evaluation’s scope and fairness.

**Ethical Concerns:**

["NO or VERY MINOR ethics concerns only"]

**Final Justification:**

The authors have clarified the concerns:
- Comparison with evidence compression literature.
- Statistical significance for provided results.
- Justification of model scales, short form QA and using GRPO.

The paper requires a major revision with a focus on using RL for evidence compression as the search/think are build on top of Search-R1 and existing literature.

**Limitations:**

Yes

**Quality:**

2

**Strengths And Weaknesses:**

### Strengths
1. The proposed method demonstrates improvements over multiple baselines, such as Search-R1 and ReSearch, by a relative margin over 20% in hard multi-hop tasks.


### Weaknesses
1. The paper appears to be an incremental development over Search-R1[1] and Search-o1[2]. It is established in the recent works, such as Search-o1[2] (Figure 2, "Get concise information and continue coherent reasoning.") and FILCO[3], that evidence compression plays critical role to avoid noise generate from imperfect retrieval.

2. Though the focus of the paper is on evidence compression, no strong baseline is chosen related to it, such as FILCO[3] and FaviComp[4]. Moreover, the Related Work section completely overlooks all evidence compression literature. Moreover, important details of the baselines are missing, such as which model size and algorithm (For Search-R1, PPO or GRPO?) was chosen for comparison.

3. Model scale and scalability: The paper does not discuss model scale’s impact on generalization or scalability. The choice of GRPO over PPO for Search-R1 is unclear, as PPO is a stronger baseline for this task in Search-R1.

4. Limited applicability: The soft-reward mechanism is restricted to short-form generations, with no exploration of its utility for long-form tasks (e.g., Self-RAG). This limits the method’s real-world relevance. It is also unclear  what is the  latency of  the method over baselines for use experience.

5. Statistical rigor: Error bars or standard deviations are omitted, such as in Table 1 and Figure  4, despite the checklist mentioning them to validate result significance. This weakens the reliability of the reported improvements.


References:

[1] Jin, Bowen, et al. "Search-r1: Training llms to reason and leverage search engines with reinforcement learning." arXiv preprint arXiv:2503.09516 (2025).

[2] Li, Xiaoxi, et al. "Search-o1: Agentic search-enhanced large reasoning models." arXiv preprint arXiv:2501.05366 (2025).

[3] Wang, Zhiruo, et al. "Learning to filter context for retrieval-augmented generation." arXiv preprint arXiv:2311.08377 (2023).

[4] Jung, Dongwon, et al. "Familiarity-aware evidence compression for retrieval augmented generation." arXiv preprint arXiv:2409.12468 (2024).

[5]  Asai, Akari, et al. "Self-rag: Learning to retrieve, generate, and critique through self-reflection." The Twelfth International Conference on Learning Representations. 2023.

---

> ### Author Rebuttal · Authors · 2025-07-30
>
> We thank the reviewer for recognizing the significant performance gain achieved in this work. We provide point-to-point responses to your primary concerns below.
>
> > W1: The paper appears to be an incremental development over Search-R1[1] and Search-o1[2]. It is established in the recent works, such as Search-o1[2] (Figure 2, "Get concise information and continue coherent reasoning.") and FILCO[3], that evidence compression plays critical role to avoid noise generate from imperfect retrieval.
>
> We appreciate the reviewer's comments on the contribution of our work, but we beg to refute the opinion that our paper is merely an incremental development.
>
> Our work addresses distinct challenges and demonstrates unique advantages:
> - **[Unique Challenges We Tackle]** We conduct **new experiments** (see Table 2 addressing Weakness 1 of reviewer KVnw) to demonstrate the challenges in developing refinement abilities via RL in Search-R1. These challenges include:
>     - Sub-optimality of simple paradigm modifications.
>     - Limited improvement from direct retrieval rewards on retrieved documents, echoing findings in previous research [1].
>     - Model benefits from non-linear retrieval reward.
> - **[Key Behaviors and Unique Benefits]** We conduct **additional experiments** (please see Table 3 addressing Weakness 1 of reviewer KVnw) to demonstrate the advantage of our method beyond simply applying summarization modules in Search-R1.
>     - No Extra Annotation Requirements, unlike previous SFT-based knowledge distillation.
>     - Recognization of missing information and adaptively plan subsequent search steps through RL (Table 5 in the paper).
>
> We hope our detailed analysis of challenges and unique benefits clearly illustrates the novel contributions of our work.
>
> > W2.1: Though the focus of the paper is on evidence compression, no strong baseline is chosen related to it, such as FILCO[3] and FaviComp[4]. Moreover, the Related Work section completely overlooks all evidence compression literature.
>
> **Response:**
> We appreciate the reviewer's suggestion for comparison with previous evidence compression methods.
>
> According to the reviewer's suggestion, we have conducted an additional experiment comparing evidence compression methods with AutoRefine comparing with (1) FaviComp [2] and (2) Search-R1 with an external summarization module [3]. The results are shown in Table 1.
>
> **Table 1:** Comparison with compression approaches.
>
> ||GeneralQA|||Multi-HopQA|||||
> |-|-|-|-|-|-|-|-|-|
> ||NQ|TriviaQA|PopQA|HotpotQA|2wiki|Musique|Bamboogle|Avg.|
> |AutoRefine|0.467|0.620|0.450|0.405|0.393|0.157|0.344|0.405|
> |FaviComp|0.302|0.502|0.410|0.240|0.220|0.054|0.232|0.280|
> |Search-R1+external refiner|0.395|0.619|0.450|0.337|0.239|0.065|0.115|0.317|
>
> The experimental results show:
> - External summarization helps on single-hop QA benchmarks (TriviaQA and PopQA);
> - AutoRefine surpasses the external summarizer in multi-hop QA setting, which highlights the unique advances summarized in the previous response addressing weakness 1.
>
> > W2.2: Moreover, important details of the baselines are missing, such as which model size and algorithm (For Search-R1, PPO or GRPO?) was chosen for comparison.
>
> **Response:**
> Thank the reviewer for pointing out the unclear implementation details of baseline methods.
> - We use GRPO as the default algorithm for RL-based baselines.
> - We use Qwen2.5-3B-Base as the default model for RL-based methods, as depicted in Sec 3.1.
>
> We hope the above clarification can address your concerns.
>
> > W3: Model scale and scalability: The paper does not discuss model scale’s impact on generalization or scalability. The choice of GRPO over PPO for Search-R1 is unclear, as PPO is a stronger baseline for this task in Search-R1.
>
> **Response:**
> We thank the reviewer for pointing out the importance of the model's scale and RL algorithm choice.
>
> **[Impact of Model's Scale]** We conduct an **additional experiment** to see the impact of the model's size, please check Table 3 in our response to Reviewer UA8w. AutoRefine surpasses previous methods by ~9% on 3B-level model and ~5% on 7B-level model.
>
> **[Choice of RL Algorithm]** We beg to refute the claim that “PPO is a stronger baseline for this task”, because:
> - As shown in Table 3 of Search-R1 [4], the GRPO algorithm has comparable performance as PPO on Qwen-3B models.
> - According to the discussion in Sec 5.1 of Search-R1 [4], "both methods GRPO and PPO) achieve similar final train reward and performance”.
>
> > W5: Limited applicability: The soft-reward mechanism is restricted to short-form generations, with no exploration of its utility for long-form tasks (e.g., Self-RAG). This limits the method’s real-world relevance. It is also unclear what is the latency of the method over baselines for use experience.
>
> **Response:**
> We thank the reviewer for this comment.
>
> **[Long Form QA Effectiveness]**
> We really appreciate the importance of long-form generation tasks, but we focuses on multi-hop QA ability which is also of high practicality. We believe that RAG models don't have to excel at every QA type to be considered contributory.
> For example, Self-RAG [5] focuses on improving the long-form QA without considering multi-hop, but it is still an excellent work with a high influence.
>
> **[Service Latency]** The additional computation of AutoRefine is merely the generation of `<refine>` content, which causes little and acceptable latency overhead. To support our claim, we showcase **additional results** about speed test between models in Table 3.
>
> **Table 3:** Speed Test of Different Methods.
> | Method| 1-by-1 Inference (sample·s) | Batched Inference (bs=512) (sample·ms) |
> |-|-|-|
> | AutoRefine| 4.743 ± 1.065| 12.921 ± 1.011|
> | Search-R1| 3.479 ± 0.542| 11.815 ± 0.980|
> | ReSearch| 3.155 ± 0.567| 11.773 ± 1.271|
>
> > W5: Statistical rigor: Error bars or standard deviations are omitted, such as in Table 1 and Figure 4, despite the checklist mentioning them to validate result significance. This weakens the reliability of the reported improvements.
>
> **Response:**
> Thanks for pointing out the importance of statistical significance. We add a **new statistical analysis** by running the experiments under different random seeds, and the results are shown in Table 4. The p-value column indicates the T-test between each baseline and AutoRefine.
>
> **Table 4:** Statistical Analysis.
> ||NQ|TriviaQA|PopQA|HotpotQA|2wiki|Musique|Bamboogle|Avg.|P-Value|
> |-|-|-|-|-|-|-|-|-|-|
> |AutoRefine|0.452 ± 0.017|0.627 ± 0.007|0.468 ± 0.017|0.423 ± 0.016|0.404 ± 0.010|0.145 ± 0.011|0.335 ± 0.023|0.408 ± 0.014|-|
> |ReSearch|0.418 ± 0.012|0.614 ± 0.014|0.451 ± 0.018|0.317 ± 0.015|0.269 ± 0.017|0.056 ± 0.015|0.132 ± 0.008|0.322 ± 0.014|5.49E-06|
> |Search-R1|0.410 ± 0.009|0.605 ± 0.019|0.429 ± 0.014|0.315 ± 0.016|0.254 ± 0.023|0.062 ± 0.005|0.127 ± 0.020|0.315 ± 0.015|2.85E-06|
>
> The results suggest the improvement achieved by AutoRefine is significant over previous search-during-think methods.
>
>
> > Q1: Reward function ablation: The reward function’s design (e.g., 0.1 as partial reward, no partial reward for correct final answers) is not analyzed. This omission limits understanding of the method’s effectiveness and trade-offs.
>
> **Response:**
> Thanks for your comment on the lack of explanation in our reward design.
> Different reward designs will indeed influence AutoRefine's performance. We conduct **additional experiments** to reveal this influence. Please refer to our Table 2 in response to Reviewer KVnw.
>
> The results indicate:
> - Retrieval reward on retrieved documents shows limited improvement. We find that directly checking whether the documents contain the answer only gives a limited performance gain. Previous researchers have also made similar conclusions [1].
> - Linear reward may over-emphasize intermediate behaviors. Adding $R_{ref}$ with $R_{ans}$ linearly as $R_{overall}=R_{ans}+R_{ref}$ is also sub-optimal. In this work, we use non-linear retrieval rewards that prioritize final answer correctness beyond merely correct knowledge refinement.
>
>
> > Q2: Hyperparameter justification: The paper lacks explicit rationale for key hyperparameters (e.g., choice of E5 as retriever, 3 documents fetched during training). If informed by prior work, this should be clearly stated to demonstrate scientific grounding.
>
> > Q3: Training configuration ambiguity: The choice of maximum search calls (5) and whether training settings align with Search-R1 are unclear. Section B.1 and Table 3 do not provide sufficient details to verify reproducibility or consistency with prior work.
>
> Thank you for the opinion. We keep the retrieval setup the same as our baseline Search-R1. We will acknowledge this inspiration in future revision.
>
> The choice of maximum search calls is empirically set, because:
> - The average search calls of our method remain less than 2.5, so no need for too many calls.
> - There’s no clear evidence that this parameter will significantly influence the result.
>
> All methods compared in Table 1 have the same experiment settings, as mentioned in Section 3.1 of our paper.
>
> We have included most of the important parameters in Section 3.1, Appendix B, and Table 3. we suggest referring to our attached code for more details.
>
>
> > Q4: Data split specification: The dev split of Hotpot-QA (distractor vs. fullwiki) is not explicitly stated, creating ambiguity about the evaluation’s scope and fairness.
>
> We use the dev split of HotpotQA, which does not contain distractor/fullwiki variants (Table 4 in paper).
>
> [1] An Empirical Study on Reinforcement Learning for Reasoning-Search Interleaved LLM Agents
>
> [2] Familiarity-Aware Evidence Compression for Retrieval-Augmented Generation
>
> [3] BART: Denoising Sequence-to-Sequence Pre-training for Natural Language Generation, Translation, and Comprehension
>
> [4] Search-r1: Training llms to reason and leverage search engines with reinforcement learning
>
> [5] Self-RAG: Learning to Retrieve, Generate, and Critique through Self-Reflection

---

> > ### Comment · Reviewer_9S2E · 2025-08-08
> >
> > I thank the authors for their rebuttals and new experiments. Though the proposed method is promising, the manuscript requires a "major revision" to better frame the necessity of RL for refinement with major focus on evidence compression. Overall, I will adjust my score to 4, voting positive. Thank you.

---

> > > ### Author Response · Authors · 2025-08-08
> > > **Reply to Comment by 9S2E**
> > >
> > > Dear Reviewer 9S2E,
> > >
> > > Thank you for your constructive comments and for adjusting your score. We appreciate your positive evaluation of our work.
> > >
> > > We are glad that our rebuttal and additional experiments have addressed your primary concerns. We fully agree with your assessment that our manuscript needs a major revision to better frame the necessity of using RL for refinement, particularly with a focus on evidence compression. The introduction and visualizations will be overhauled in our next revision to clearly demonstrate the key challenges and benefits of our RL-driven knowledge refinement approach.
> > >
> > > Best regards,
> > >
> > > The Authors

---

> > > ### Author Response · Authors · 2025-08-09
> > >
> > > Dear Reviewer 9S2E,
> > >
> > > Thank you for your recognition of our work. We're pleased that our rebuttal and new experiments addressed your concerns, and we appreciate your decision to adjust your score.
> > >
> > > We've noticed that the score in the system still reflects a 2. Since these platforms can sometimes have a delay and the discussion deadline is approaching, we would be very grateful if you could quickly confirm that your updated score of 4 was successfully saved.
> > >
> > > Best regards,
> > >
> > > The Authors

---

> ### Author Response · Authors · 2025-08-03
> **Looking forward to your reply**
>
> Dear Reviewer 9S2E,
>
> Thank you again for your constructive suggestions. We have carefully addressed your concerns by:
> - elaborating on distinct challenges and unique advantages in this research comparing with Search-R1
> - including new baseline methods and clarifying baseline/hyperparam settings
> - demonstrating the acceptable latency overhead and scalability of our method
> - adding statistical analysis
> - disambiguating data split and training configurations
>
> We look forward to further discussion with you and would greatly appreciate your positive feedback on our rebuttal.
>
> Best regards,
>
> The Authors

---

> ### Author Response · Authors · 2025-08-06
> **New experimental results & Looking forward to your reply**
>
> Dear Reviewer 9S2E,
>
> Regarding your feedback on the evidence compression baseline, we are pleased to share the new experimental results we obtained during the discussion period.
>
> > W2.1: Though the focus of the paper is on evidence compression, no strong baseline is chosen related to it, such as FILCO[3] and FaviComp[4]. Moreover, the Related Work section completely overlooks all evidence compression literature.
>
> In our rebuttal, we compared FaviComp and Search-R1 + BART as baselines. However, we realized that the BART model might be considered small and somewhat outdated, so we supplemented our experiments using the Qwen2.5 model as an external refiner for Search-R1. Specifically, we prompted Qwen2.5-3B-Instruct to perform evidence compression in two settings:
> 1. Ask the model to summarize the documents
> 2. Ask the model to summarize the documents, and make plans for the next search step
>
> The results are presented in the table below.
>
> Table: Performance Comparison with different external refiners
>
> |                                                                               | General QA |          |       | Multi-Hop QA |       |         |           |       |
> | ----------------------------------------------------------------------------- | ---------- | -------- | ----- | ------------ | ----- | ------- | --------- | ----- |
> |                                                                               | NQ         | TriviaQA | PopQA | HotpotQA     | 2wiki | Musique | Bamboogle | Avg.  |
> | AutoRefine                                                                    | **0.467**      | **0.620**    | **0.450** | **0.405**        | **0.393** | **0.157**   | **0.344**     | **0.405** |
> | FaviComp                                                                      | 0.302      | 0.502    | 0.410 | 0.240        | 0.220 | 0.054   | 0.232     | 0.280 |
> | Search-R1 + external refiner (BART)                                           | 0.395      | 0.619    | **0.450** | 0.337        | 0.239 | 0.065   | 0.115     | 0.317 |
> | Search-R1 + external refiner (Qwen2.5-3B-Instruct, summary)                   | 0.399      | 0.600    | 0.445 | 0.331        | 0.264 | 0.073   | 0.180     | 0.328 |
> | Search-R1 + external refiner (Qwen2.5-3B-Instruct, summary & plan) | 0.378      | 0.562    | 0.431 | 0.299        | 0.231 | 0.059   | 0.149     | 0.301 |
>
> From these new results, we find that:
> - As expected, the newer and larger Qwen2.5-3B-Instruct serves as a better summarization model than BART (a 0.011 overall improvement).
> - Autorefine demonstrates superior performance compared with the evidence compression baselines.
>
> We hope these results better address your concerns.
>
> Besides, we want to kindly remind you that the discussion period is coming to a close. We look forward to hearing from you.
>
> Best regards,
>
> The Authors

---

> ### Author Response · Authors · 2025-08-07
> **Look forward to hear from you**
>
> Dear Reviewer 9S2E,
>
> Thank you for your questions regarding our model's scalability, service latency, statistical rigor, and missing of baseline methods. To address your concerns, we've conducted additional comparisons at the 7B level, a latency test, statistical analysis, and added new baseline methods. We hope our response has resolved your main concerns and we would appreciate it if you would consider increasing our score. If you still have some concerns, we would be very happy to have the opportunity to continue our discussion with you.
>
> Best regards,
>
> The Authors

---

### Official Review · Reviewer_jRtt · 2025-07-05

**Clarity:** 4
**Significance:** 2
**Originality:** 3
**Rating:** 5
**Confidence:** 4

**Summary:**

The paper introduces AutoRefine, an RL post-training framework that RAG by adding an explicit “search-and-refine-during-think” reasoning loop. In each reasoning cycle the model (i) plans (\<think\>), (ii) issues a search query (\<search\>), (iii) receives documents, (iv) refines those documents to distill the most relevant facts (\<refine\>), and (v) eventually produces an answer (\<answer\>). In this reasoning process, the authors introduced two rewards: 1) the outcome reward (whether the final answer is correct) and 2) the retrieval reward, which is measured based on the quality of refined documents within \<refine\>\</refine\> blocks. The model is trained end2end using GRPO. To demonstrate the effectiveness of the proposed method, the authors conducted experiments across a set of sing-hop and multi-hop datasets. The strong empirical results show that AutoRefine is effective in identifying and addressing knowledge gaps via multi-turn, high-quality search queries.

**Questions:**

n/a

**Ethical Concerns:**

["NO or VERY MINOR ethics concerns only"]

**Final Justification:**

The added experiments resolved some of my major concerns. I think adding these results could significantly help the audience understand the effectiveness of the framework design. Therefore, I am willing to increase my score to 5.

**Limitations:**

yes

**Quality:**

3

**Strengths And Weaknesses:**

### Strengths

- The proposed "search-and-refine-during-think" approach is technically sound and well-presented.
- Strong empirical results over a set of single-hop and multi-hop datasets. It not only outperforms the baselines but also the strong results in the literature.
- The experiments are comprehensive, including thorough ablations showing the effectiveness of the framework design.

### Weaknesses

The major concern I have regarding this paper is its similarity to the Search-R1 paper. The core architecture, prompt template, RL framework, datasets, and evaluation metrics of AutoRefine mirror those of Search-R1. The new elements introduced in AutoRefine are (i) inserting a <refine> block in which the model summarizes the retrieved passages and (ii) adding a “retrieval-specific” reward that checks whether the gold answer appears inside that block. Search-R1 could adopt these two tweaks with minimal changes. Although I admit that making those two changes brings significant gains over multi-hop reasoning datasets, I am not fully convinced that it is worth a NeurIPS publication. From my perspective, exploring more diverse experimental settings could make the paper much stronger. For example, if the ground-truth is given as a long and complex answer, how to compute the retrieval reward, and how does that affect the final performance?

---

> ### Author Rebuttal · Authors · 2025-07-30
>
> We sincerely appreciate the reviewer's acknowledgement of the soundness, strong performance and comprehensive analysis. Below are point-to-point responses addressing your major concerns.
>
> > **Weakness 1: Similarity to the Search-R1 paper.** Search-R1 could adopt these two tweaks with minimal changes. I’m not fully convinced that it is worth a NeurIPS publication.
>
> **Response:**
> Thank you for your constructive feedback. We understand your concern regarding the similarity to Search-R1 and the novelty of our contribution.
>
> While AutoRefine builds upon the foundational search-during-reasoning paradigm established by Search-R1, we respectfully disagree that our method constitutes "two tweaks with minimal changes." Here we would like to clarify the **key challenges** we tackled and the **key advances** we achieved in our paper:
>
> **[Key Challenges]** To demonstrate the key challenges in developing knowledge refinement steps in search-during-think methods, we conduct **additional experiments** comparing different supervision reward signals, as shown in Table 1. We empirically show that:
> - **Simple paradigm modification (e.g., "search-and-refine-during-think") is sub-optimal.** As observed in Table 1, merely adding a refinement step does improve the overall performance. Nevertheless, we uncover that the model's refinement ability can be further enhanced with proper retrieval-related rewards.
> - **Direct retrieval reward on retrieved documents showcases limited improvement.** One simple way to add a retrieval-related signal is to check the existence of the final answer in the retrieved documents. Nevertheless, as shown in Table 1, we find that this improvement is also not the ultimate solution. Previous researchers have made similar conclusions: direct retrieval reward can not always improve the performance of search-during-thinking models [1].
> - **Linear rewards may over-emphasize intermediate behaviors.** Our findings clearly indicate that a linear combination of answer and refinement rewards ($R_{overall}$​=$R_{ans}$​+$R_{ref}$​) is _inferior_ to our proposed non-linear reward design, which prioritizes the final answer correctness while still fostering robust refinement capabilities. This intricate balance in the reward function is a core innovation of AutoRefine, directly contributing to its superior performance across various QA benchmarks.
>
> **Table 1:** Comparison Between Different Rewards Used in AutoRefine.
>
> |                                            | General QA |           |           | Multi-Hop QA |           |           |           |           |
> | ------------------------------------------ | ---------- | --------- | --------- | ------------ | --------- | --------- | --------- | --------- |
> |                                            | NQ         | TriviaQA  | PopQA     | HotpotQA     | 2wiki     | Musique   | Bamboogle | Avg.      |
> | AutoRefine-Reward at Refine - nonlinear    | **0.467**  | **0.620** | **0.450** | **0.405**    | **0.393** | **0.157** | **0.344** | **0.405** |
> | AutoRefine-Reward at Refine - linear       | 0.415      | 0.593     | 0.435     | 0.376        | 0.365     | 0.143     | 0.296     | 0.375     |
> | AutoRefine-Reward at Documents - nonlinear | 0.418      | 0.592     | 0.441     | 0.381        | 0.386     | 0.153     | 0.320     | 0.384     |
> | AutoRefine-Reward at Documents - linear    | 0.417      | 0.590     | 0.414     | 0.387        | 0.360     | 0.152     | 0.304     | 0.375     |
> | AutoRefine-only answer reward              | 0.423      | 0.583     | 0.424     | 0.368        | 0.351     | 0.139     | 0.344     | 0.371     |
> | Search-R1                                  | 0.421      | 0.583     | 0.413     | 0.297        | 0.274     | 0.066     | 0.128     | 0.312     |
>
>
> **[Key Advances]** Building on our solutions to these challenges, AutoRefine offers key advances that enhance performance and capabilities, such as:
> - **Summarization ability with no data annotation.** AutoRefine successfully activates the intrinsic knowledge refinement ability embedded within LLMs, resulting in significant performance gain across single- and multi-hop benchmarks.
> - **Introspection of missing knowledge.** In multi-hop search-during-think methods, mere summarization may not be sufficient. The model also needs to recognize missing information and adaptively plan subsequent search steps. We observed that AutoRefine successfully developed this capability in the refinement steps through RL (Table 5 in our paper), which significantly contributes to our superior performance on multi-hop benchmarks.
>
> Upon your opinions, we have acknowledged that our delivery in the paper does not fully reveal the challenges in developing the spontaneous refinement ability and unique advances of our method. We hope our explanation has successfully addressed your concerns regarding the novelty and contribution of this work.
>
>
> > **Weakness 2: More experimental settings** exploring more diverse experimental settings could make the paper much stronger. For example, if the ground-truth is given as a long and complex answer, how do we compute the retrieval reward, and how does that affect the final performance?
>
> **Response:**
> Thank you for this suggestion. We entirely agree that exploring more diverse experimental settings, especially concerning the nature of the ground-truth answers and their impact on `retrieval reward` computation, is crucial for strengthening our paper. Besides the experiment added in Table 1, we have conducted **additional experiments** that explore how a more fine-grained retrieval reward would affect the model's performance on long and complex answers.
>
> **[Adapting Retrieval Rewards for Complex Answers]** Except for the existing cover-exact match used in our paper, we've explored two additional reward designs:
> 1. Token-level recall: Calculates how many *tokens* in the ground-truth answer appear in the refined documents,
> 2. Word-level recall: Calculates how many *words* in the ground-truth answer appear in the refined documents;
>
> under different benchmark settings:
> - Full dataset: Test on all questions in the benchmark.
> - Complex answers: Test on questions with an answer >5 words.
>
> The results are shown in Table 2.
>
> **Table 2:** Comparison between original and finer-grained retrieval reward design.
>
> |                                                  | **General QA** |           |           | **Multi-Hop QA** |           |           |
> | ------------------------------------------------ | -------------- | --------- | --------- | ---------------- | --------- | --------- |
> |                                                  | TriviaQA       | PopQA     | HotpotQA  | 2wiki            | Musique   | Avg.      |
> | **Full Dataset**                                 |                |           |           |                  |           |           |
> | AutoRefine - CEM Retrieval Reward                | **0.620**      | **0.450** | **0.405** | 0.393            | **0.157** | **0.405** |
> | AutoRefine - Token-level Recall Retrieval Reward | 0.604          | 0.433     | 0.376     | 0.364            | 0.136     | 0.383     |
> | AutoRefine - Word-level Recall Retrieval Reward  | 0.609          | 0.437     | 0.395     | **0.395**        | 0.142     | 0.396     |
> | **Complex Answers  (>5 words)**                  |                |           |           |                  |           |           |
> | AutoRefine - CEM Retrieval Reward                | 0.128          | 0.261     | 0.105     | 0.368            | 0.023     | 0.177     |
> | AutoRefine - Token-level Recall Retrieval Reward | **0.132**      | 0.292     | 0.094     | 0.379            | 0.047     | 0.189     |
> | AutoRefine - Word-level Recall Retrieval Reward  | 0.131          | **0.375** | **0.113** | **0.409**        | **0.054** | **0.216** |
>
> The results presented in Table 2 confirm that a more fine-grained retrieval reward can greatly improve the model's performance when answers have complex forms, while keeping the overall performance approximately the same.
>
> We hope these additional results could help strengthen our experiment design.
>
> [1] Jin, Bowen, et al. "An Empirical Study on Reinforcement Learning for Reasoning-Search Interleaved LLM Agents." arXiv preprint arXiv:2505.15117 (2025).

---

> > ### Comment · Reviewer_jRtt · 2025-08-08
> > **Thanks for the experiments**
> >
> > Thanks for the added experiments which resolved some of my major concerns. I think adding these results could significantly help the audience understand the effectiveness of the framework design. Therefore, I am willing to increase my score to 5. One more suggestion: it would be very interesting to see how the proposed framework does on long-form answers, as the retrieval signals will be hard to eval. It can be heuristic-based score or model-based score, and I am quite curious to see the extent to which it helps the full system.

---

> ### Author Response · Authors · 2025-08-03
> **Looking forward to your reply**
>
> Dear Reviewer jRtt,
>
> Thank you again for your constructive opinions. We have carefully addressed your concerns by:
> - elaborating on the key challenges we tackled and the key benefits we achieved in this research
> - exploring more diverse experimental settings on long and complex QA
>
> We look forward to further discussion with you and would greatly appreciate your positive feedback on our rebuttal.
>
> Best regards,
>
> The Authors

---

### Note · Authors · 2025-08-12

Dear ACs/SACs/PCs,

We would like to summarize the strengths of this work acknowledged by the reviewers, and our responses to address all the reviewers’ concerns.

---

The reviewers highlighted the following **strengths of our paper**:
1. **Novelty:** novel refine step, introduce new elements (Reviewer $\textcolor{blue}{KVnw}$, $\textcolor{green}{jRtt}$)
2. **Well-organized:** sound and well-presented (Reviewer $\textcolor{green}{jRtt}$), well-written and easy to follow (Reviewer $\textcolor{blue}{KVnw}$), well-chosen and complete experiments (Reviewer $\textcolor{red}{UA8w}$)
3. **Strong Performance:** simple yet effective (Reviewer $\textcolor{blue}{KVnw}$), strong for multi-hop QA (Reviewer $\textcolor{red}{UA8w}$), over 20% improvements in hard multi-hop tasks (Reviewer $\textcolor{orange}{9S2E}$, $\textcolor{green}{jRtt}$)

---

The **common concerns** raised by the reviewers were:
1. The uniqueness of our method vs. prior work (Reviewer $\textcolor{green}{jRtt}$, $\textcolor{orange}{9S2E}$, $\textcolor{blue}{KVnw}$)
2. Scalability on models with $\geqslant$ 3B parameters (Reviewer $\textcolor{orange}{9S2E}$, $\textcolor{red}{UA8w}$)
3. More explorations over proposed retrieval reward (Reviewer $\textcolor{green}{jRtt}$, $\textcolor{red}{UA8w}$)
4. Inclusion of baseline about evidence compression (Reviewer $\textcolor{orange}{9S2E}$, $\textcolor{blue}{KVnw}$)

Our **responses** to these concerns are summarized below:
1. We have **elaborated on the unique challenges and benefits** of our work. Additionally, we've included a summary table detailing the technical advancements of our method compared to previous research.
2. We have **extended our method to 7B-level models**, demonstrating consistent performance gains at different model scales.
3. We've conducted new experiments with **six additional retrieval reward setups**. Furthermore, we've included a thorough analysis of how reward designs influence model behavior across different situations (single-hop/multi-hop/complex answers).
5. We have introduced **two additional baseline methods** (one of them under three different settings) that utilize evidence compression techniques in RAG settings.

All major concerns have been successfully addressed, and we are grateful for the positive feedback from the reviewers. We hope that the improvements made will be taken into consideration.

We sincerely appreciate your valuable time and patience!

Thanks and regards,

Authors

---

### Decision · Program_Chairs · 2025-09-17

**Decision:**

Accept (poster)

**Comment:**

The paper attack an important problem, how to reduce cost of LLM annotation without losing much accuracy. The method is simple but novel and experiments are strong. In my view it is solid and useful, so I support accept.